# Re-assessment of net energy production and greenhouse gas emissions avoidance after 40 years of photovoltaics development

Atse Louwen[1], Wilfried G.J.H.M. van Sark[1], André P.C. Faaij[2] & Ruud E.I. Schropp[3]

Since the 1970s, installed solar photovoltaic capacity has grown tremendously to 230 gigawatt worldwide in 2015, with a growth rate between 1975 and 2015 of 45%. This rapid growth has led to concerns regarding the energy consumption and greenhouse gas emissions of photovoltaics production. We present a review of 40 years of photovoltaics development, analysing the development of energy demand and greenhouse gas emissions associated with photovoltaics production. Here we show strong downward trends of environmental impact of photovoltaics production, following the experience curve law. For every doubling of installed photovoltaic capacity, energy use decreases by 13 and 12% and greenhouse gas footprints by 17 and 24%, for poly- and monocrystalline based photovoltaic systems, respectively. As a result, we show a break-even between the cumulative disadvantages and benefits of photovoltaics, for both energy use and greenhouse gas emissions, occurs between 1997 and 2018, depending on photovoltaic performance and model uncertainties.

[1] Copernicus Institute of Sustainable Development, Utrecht University, Heidelberglaan 2, 3584 CS Utrecht, the Netherlands. [2] Energy & Sustainability Research Institute, University of Groningen, Blauwborgje 6, P.O.Box 9700 AE Groningen, the Netherlands. [3] Plasma & Material Processing, Department of Applied Physics, Eindhoven University of Technology (TU/e), P.O. Box 513, 5600MB Eindhoven, the Netherlands. Correspondence and requests for materials should be addressed to A.L. (email: a.louwen@uu.nl) or to W.G.J.H.M.v.S (email:w.g.j.h.m.vansark@uu.nl).

Cumulative installed solar photovoltaic (PV) capacity (CIPC) grew from less than 1 MW$_p$ in 1975 to around 180 GW$_p$ at the end of 2014 (refs 1–3), with a compound annual growth rate (CAGR) of 45%. As shown in Fig. 1, major installation markets at the beginning of the 1990s were Japan and Italy, but from 2005 to 2014 Germany was the leading PV market in terms of CIPC[4]. It is expected that China will surpass Germany as the country with the largest CIPC during 2015[5]. The strong growth can largely be attributed to successful government support schemes, like Germany's feed-in tariff, but also to rapidly falling prices of PV systems.

PV electricity has large social and governmental support, as during its operation no harmful emissions are released. Over the whole life-cycle of a PV system, it pays back the energy invested and greenhouse gas (GHG) emissions released during its production multiple times[6–9]. As PV systems operate over a period of up to 30 years, there is a significant time-lag between the investments, in terms of cumulative energy demand (CED) and GHG emissions, and the benefits obtained due to delivery of electricity and replacement of high-environmental impact electricity from fossil fuel sources. Coupling the rapid growth of PV with this context of upfront investments has led to some concerns, regarding the PV industry's environmental sustainability. A fast growth of installed PV capacity could result in the creation of an energy sink, as the PV industry could embed energy in PV systems at a rate outpaced by these system's ability to deliver it back. The same can be true for GHG emissions, when the production of PV systems releases more GHG emissions than the electricity produced with PV can offset by replacing more GHG intensive electricity. Although there is evidence that shows that CED and GHG emissions are correlated[10], this is not necessarily the case.

To avoid the creation of an energy and/or GHG sink, in general, the growth of the industry should be limited by 1/PBT[11,12], where PBT (payback time) is the time in which upfront investments in either CED or GHG emissions are paid back. However, energy and GHG sinks from periods of growth exceeding 1/PBT can be offset by decreased growth rates (or decreasing PBT) in later stages. Thus, the dynamics of growth need to be taken into account, rather than always aiming for a 1/PBT limited growth, as is discussed by Emmott et al.[13] The concept of the PV industry as an energy sink, and more recently GHG sink is well known in the PV community. Grimmer et al.[11] have been one of the first to address this issue in terms of energy, stating that to maximise the (positive) impact of solar technologies, they should have short energy payback time

(EPBT) and long lifetime. When the growth of the PV industry started to accelerate, others indicated the necessity of strong decreases in energy payback time[12]. Others have also analysed the relation between industry growth and EPBT and concluded that for mono-crystalline PV, a sustainable growth rate should be limited to around 7% (ref. 14), however this result was based on a static measurement of the energy footprint of PV systems, and thus did not account for the decrease of the energy footprint of PV systems over time. More recent studies have also analysed GHG sinks[13,15].

Here, we review the development of environmental impact of production of PV systems over time, focusing on greenhouse gas emission and energy demand, and analysing only mono- and polycrystalline silicon based systems, as these cover over 90% of total installed capacity[16]. We gather results from a total of 40 life-cycle assessment (LCA) studies of PV systems (including inverters and support structure) conducted from 1976 to 2014, and couple these results to development of cumulative installed capacity figures, to show the development of energy demand and greenhouse gas emissions from PV production as a function of installed capacity, and to establish experience curves and learning rates[17] for these parameters. The models obtained are used in conjunction with scenarios on performance of PV in order to calculate net contributions of the PV industry as a whole, in terms of energy and greenhouse gas emissions, and to calculate when break-even between environmental investments and benefits occurs. A similar approach was used before by Dale and Benson[18], who focused on net electricity, and analysed studies in a narrower timeframe between 1990 and 2010. The authors concluded that cumulative break-even will occur somewhere before 2020. Other studies have focused on GHG emissions[13] even taking into account the gradual effect of GHG emissions on radiative forcing[15]. The latter two studies however focused on case studies or PV installation targets. Here, we want to combine approaches, by taking into account actual PV industry growth, and analysing the environmental impact using LCA studies from a wider time period. We show that there are strong downward trends for both energy demand and GHG emission from PV production, and that these trends follow the experience curve law. For every doubling of installed PV capacity, there is a decrease in energy use of 13% and 12% and in greenhouse gas footprint of production of 17 and 24%, for poly and monocrystalline based PV systems, respectively. As a result, there is a break-even since 2011 between the cumulative detriments and benefits of PV, in terms of both energy use and greenhouse gas emissions for a scenario that takes into account

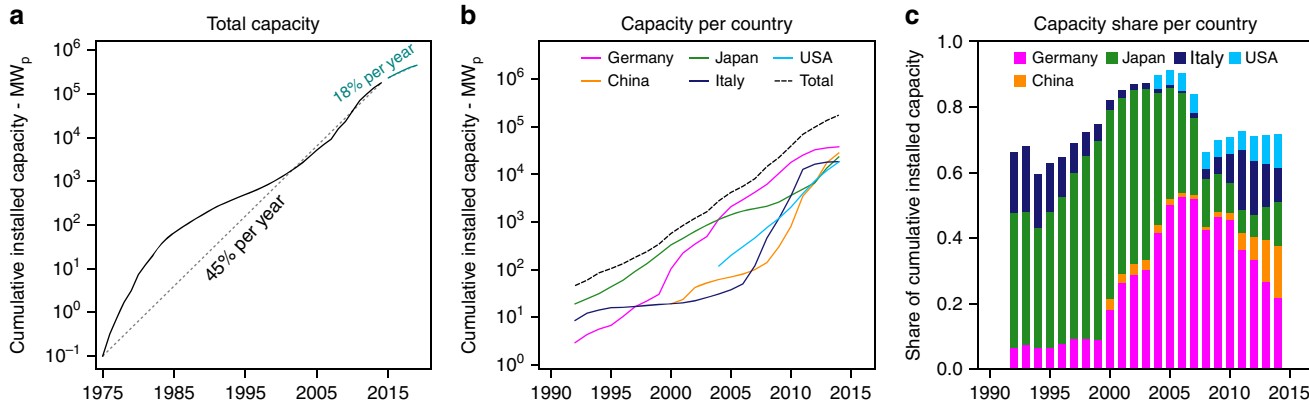

**Figure 1 | Historical PV market developments.** (**a**) Development of total Cumulative Installed PV Capacity (all PV technologies) from 1975–2014 with a CAGR of 45%; data taken from[1–3,16,25,46], and expected development from 2015–2020 (CAGR: 18%[1],). (**b**) Development of CIPC from 1992–2014 for 5 main markets; data taken from[2,46]. (**c**) Development of total capacity share from 1993–2014 for 5 main markets; data taken from[2,46].

PV production location over time and a realistic PV performance scenario. Taking into account a worst-case PV performance scenario and model uncertainties, break even occurs in 2017 for net energy, and in 2018 for greenhouse gas emissions avoidance.

## Results

**Historical development of cost and environmental footprint.** The development of cost and environmental footprint over the period 1975–2015 is shown in Fig. 2. Within this period, with an installed capacity increase from less than 1 MW$_p$ to almost 180 GW$_p$, prices dropped from almost 100 USD per W$_p$ to around 0.64–0.67 USD per W$_p$ at the end of 2014. Data for environmental footprint of PV systems do not go back that far, and furthermore show a less clear trend over time. Still, especially for energy pay-back time (which is calculated from reported system CED according to the procedure described in the Methods section) a clear decrease of environmental footprint over time can be observed. Energy pay-back times drop from around 5 years in 1992 to around just under 1 year for poly-Si and just over 1 year for mono-Si PV systems currently[8]. Greenhouse gas emissions from photovoltaics, expressed in grams of $CO_2$-equivalent per kilowatt-hour (gCO$_2$-eq kWh$^{-1}$), show large variations, even for studies analysing PV systems from the same year. This large variation is likely due to a variety of approaches in calculating the GHG emissions. More recent studies seem to use more congruent methods, resulting in a smaller variation of calculated GHG footprint. The current GHG footprint (harmonized data) is around 20 gCO$_2$-eq kWh$^{-1}$ for poly-Si PV systems, and around 25 gCO$_2$-eq kWh$^{-1}$ for mono-Si PV systems[8], down from 143 gCO$_2$-eq kWh$^{-1}$ for poly-Si in 1992 (ref. 19) and 409 gCO$_2$-eq kWh$^{-1}$ for mono-Si in 1986 (ref. 20). For determining the EPBT and the GHG footprint per kWh of produced electricity, the energy yield of the systems, and thus insolation and consequently location, are of great importance. The values here refer to standardised conditions: insolation of 1,700 kWh m$^{-1}$y$^{-1}$ and a performance ratio (PR) of 0.75, based on methodology guidelines from[21]. Recent meta-analyses of LCA studies on crystalline PV systems established average values for environmental footprint of PV systems, and found energy payback times to be 3.1 and 4.1 years for poly and mono-Si, respectively[22], based on studies conducted between 2005 and 2013. In a recent study Ferroni and Hopkirk[23] presented figures for energy return on energy invested that are equivalent to energy payback times that were much higher compared to what is found in other recent studies, or even much older studies. The study was

not well received within the PV research community (M. Raugei, personal communication), and was found to severely lack in the applied methodology. For instance, the authors strongly overestimated the energy required for PV production, and at the same time underestimated the energy yield and lifetime of PV systems. Another meta-analysis found GHG emissions to be 44.3 and 79.5 gCO$_2$-eq kWh$^{-1}$ for poly and mono-Si, respectively, based on studies from 2004 to 2014 (ref. 24). Comparing this to the values mentioned earlier, that were reported for current PV systems by Wetzel[8], we see that the developments in the past decade have been so significant that the GHG footprint of state-of-the-art poly and monocrystalline silicon based PV systems is already 55 to 69% lower, respectively, compared to the averages of roughly the last decade (2004–2014) mentioned at the beginning of this paragraph.

**Learning rates.** Figure 3 shows the data we have obtained for the average selling price of PV modules, and CED and GHG footprint of mono- and polycrystalline silicon-based PV systems, including inverter and support structure. It also shows the fitted experience-curve models, and an uncertainty range around this fitted model. As was established before[16,25,26], there is a very clear correlation between the price of PV and the cumulative PV production. In the price-experience curve in Fig. 3 we can observe two events: First, a price plateau appears when installed capacity was around 10 GW$_p$, due to a polysilicon feedstock supply shortage around 2008, and secondly, a strong drop of price is observed after this plateau to levels below the learning curve, due to oversupply of polysilicon and PV modules resulting from production capacity expansions. From 2012 onwards, prices have stabilised and are thus returning to values projected by the learning curve. Over the whole period, a learning rate of 20.1 ± 0.5% can be observed (error bars refer to the standard deviation of the fitted parameter).

For CED, both technologies show a downward trend of CED *versus* installed capacity, with learning rates of 12.6 ± 0.85% and 11.9 ± 1.04% for poly and mono-Si systems. An earlier study indicates learning rates specific for rooftop and ground-mounted PV systems, a distinction not made in our analysis, and reports learning rates of 13 and 11% for ground mounted and 18 and 14% for rooftop-mounted poly and monocrystalline silicon based PV systems, respectively[27].

The quality of our fit is lower for CED compared to that for cost, likely due to a variety of methods employed to calculate the CED, such as different system boundaries or assumptions of energy usage during PV production. Especially over time

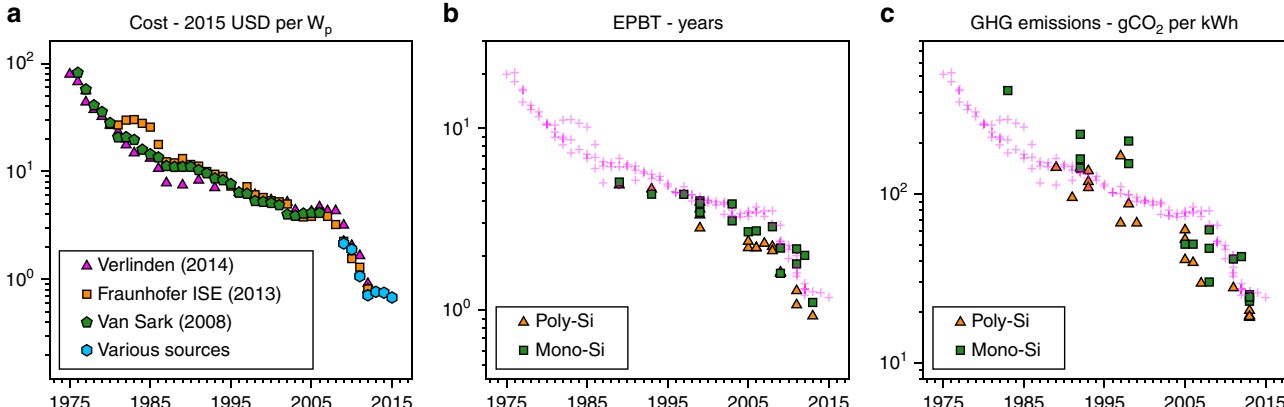

**Figure 2 | Development of cost and environmental impact of PV.** (**a**) Development of average module selling price over time, in 2015 USD per W$_p$. Data from[16,25,26,34,35]. (**b**) Development of energy payback time over time. (**c**) Development of greenhouse gas emissions from PV electricity over time. The magenta crosses in (**b**,**c**) are an overlay of the cost data from (**a**).

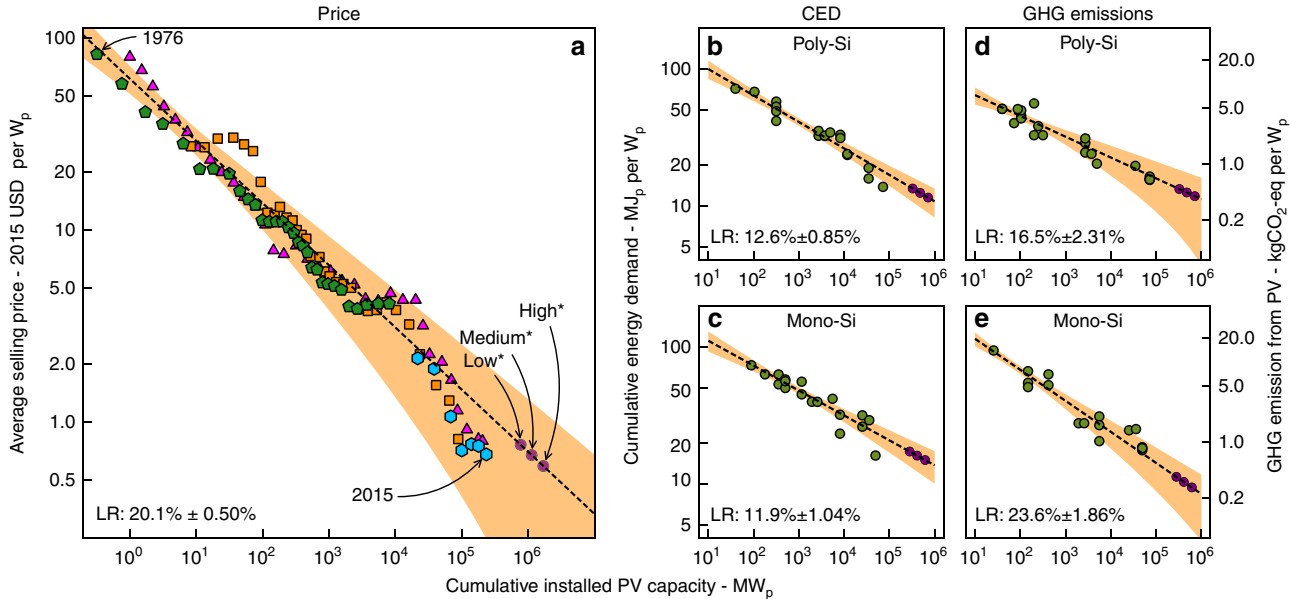

**Figure 3 | Experience curves for cost and environmental footprint of mono- and polycrystalline PV modules and systems.** (**a**) Experience curve for average module selling price. Magenta triangles: data from[26]; Orange squares: data from[16], Green pentagons: data from[25]; Blue hexagons: data from[34,35]. (**b–e**) Experience curves for cumulative energy demand and GHG emissions from production of mono- and polycrystalline silicon based PV systems. The purple circles indicate the predicted cost, based on forecasted values of cumulative installed PV capacity for the year 2040. These values were obtained by adjusting the starting points of the 2014 IEA 'World Energy Outlook'[52] long term scenarios to the more recent 'Global Market Outlook for Photovoltaics 2015–2019' short-term scenarios[1]. (**a–e**) The experience curves are indicated by the dashed black lines. The shaded areas show a 95% confidence interval for the fitted models. (**d,e**) The graphs for GHG emission show data harmonised for lifetime and annual yield and degradation (right) as described in the Methods section.

these methods likely have changed considerably. However, many of the datapoints for CED also do not necessarily reflect a market average, contrary to the price data, but are often studies on specific producers, thus there is no convergence of the CED like there is for price in a globally operating PV market.

For greenhouse gas emissions from PV production, we also observe an even clearer downward trend for both technologies with increasing installed capacity, with learning rates of $16.5 \pm 2.31\%$ and $23.6 \pm 1.86\%$ for poly- and monocrystalline based systems respectively. As reflected by the higher errors in the learning rates and wider confidence intervals, the quality of the fit is somewhat lower compared to that for CED and especially cost, and would likely benefit from data that is more evenly spread over time. Especially for poly-Si, a large number of studies was performed between 1995 and 1999, after which there is a gap in the data between up to 2005. A more even dispersement of studies over a longer period of time would likely result in higher fit quality and lower parameter error. We observe a stronger learning rate for mono- compared to polycrystalline silicon-based PV systems. This is likely due to the fact that mono-crystalline silicon PV module production is more energy intensive, and thus benefits not only from energy usage reduction, but also, more than poly-Si, from reduction of the GHG footprint of energy used as input in the production processes, that occurs independently over time as show in the data from the UN[28] and the World Resources Institute (http://cait.wri.org).

**Outlook for cost and environmental impact.** The future development of PV is very difficult to predict, as every year of development seems to exceed our expectations. Forecasts for installed capacity in 2040 by the IEA in their World Energy Outlook 2014 range from $0.6$–$1.4\,\mathrm{TW_p}$, figures we have adjusted to $0.7$–$1.6\,\mathrm{TW_p}$ based on more recent developments of installed capacity and short term forecasts[1]. Extrapolating the learning curves obtained from historical data to these cumulative PV

capacity values, PV module costs are calculated to be 0.59 to 0.76 (2015 USD) $\mathrm{W_p^{-1}}$, of which the high end of the range is actually well above current module factory gate selling prices of 0.68 USD $\mathrm{W_p^{-1}}$ for crystalline silicon modules[29]. This seems to confirm that current module prices are quite far below what we would have expected from about 40 years of price development.

For environmental impact, cumulative energy demand drops to $11.5$–$13.3\,\mathrm{MJ_p\,W_p^{-1}}$ for poly and $14.9$–$17.2\,\mathrm{MJ_p\,W_p^{-1}}$ for mono-crystalline based PV systems, equating to energy pay-back times of $0.8$–$0.9$ and $1.0$–$1.2$ years, respectively. Greenhouse gas emissions are extrapolated to drop to $0.40$–$0.49$ and $0.27$–$0.37\,\mathrm{kgCO_2}$-eq $\mathrm{W_p^{-1}}$ or $12$–$14$ and $8.0$–$11\,\mathrm{gCO_2}$-eq $\mathrm{kWh^{-1}}$ for poly- and monocrystalline systems, respectively. Note that in this latter case monocrystalline systems actually have lower environmental impact compared to polycrystalline systems, contrary to what is observed presently, due to the higher learning rate calculated. The projections indicate that in order to make such low GHG footprints possible, there is likely a need for a strong reduction of the GHG intensity of the energy inputs of PV production. Although many governments have put into place targets to achieve this, it remains to be seen if this will be the case.

**Net societal contributions of PV.** Figure 4 shows the development of net energy use and net avoided emissions, respectively, for the two PV performance scenarios, and for three production location scenarios (see Methods section). The solid lines show results for the 'Increasing $PR$' scenario, while the dashed lines show results for the 'Low $PR$' scenario. In the 'Increasing $PR$' scenario, $PR$ of PV systems increases over time from 1975 to 2015 and remains constant thereafter, while in the 'Low $PR$' scenario, we assume a constant, worst-case $PR$ (see Methods section). Figure 4a shows that a break even in terms of net primary energy has likely already occurred. Even for the 'Low $PR$' scenario,

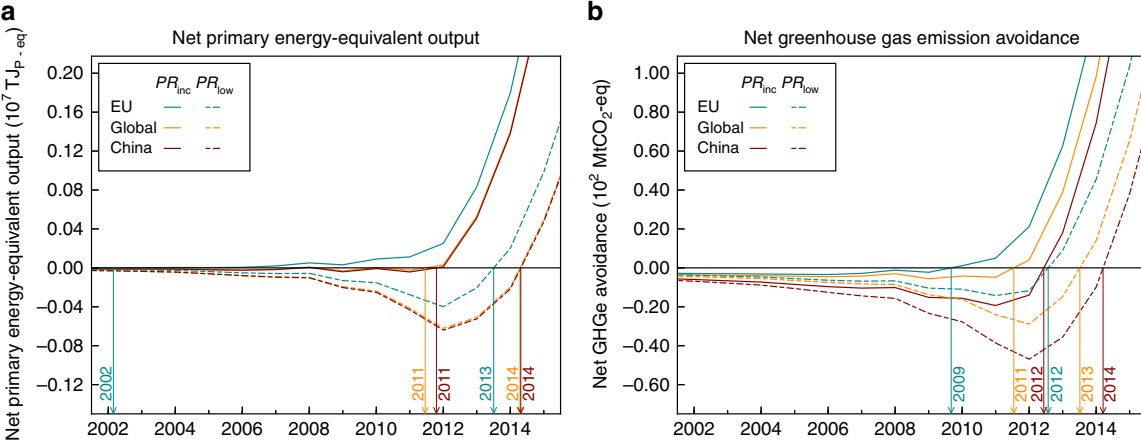

**Figure 4 | Calculated cumulative net environmental impact of cumulative installed PV capacity.** (**a**) Cumulative net energy output, in terajoules of primary energy equivalent. (**b**) Cumulative net greenhouse gas emissions avoidance, in megatonnes (Mt) of $CO_2$-equivalent. Results are shown for 2 PV performance scenario's: 'Increasing $PR$' and 'Low $PR$', and 3 production location scenarios (see Methods section).

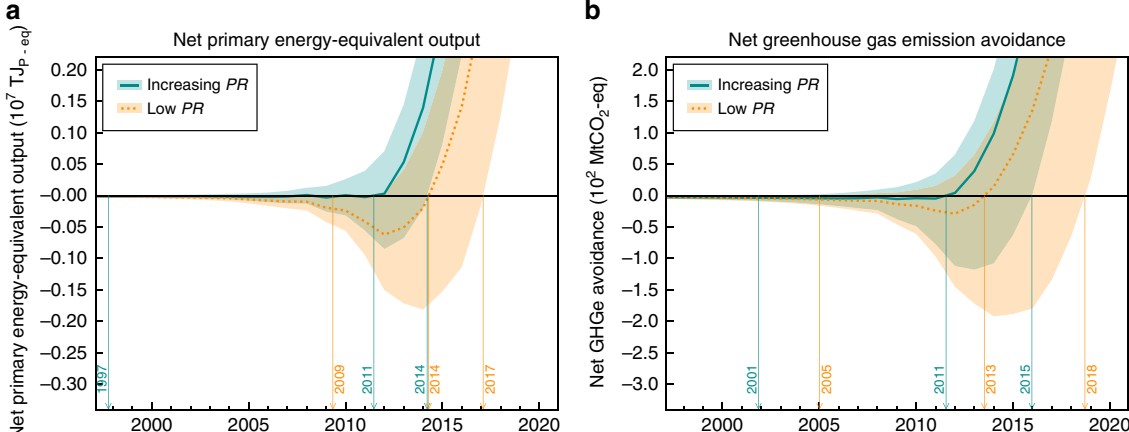

**Figure 5 | Results of the Monte Carlo uncertainty analysis on the net contribution of PV.** Results are shown for the 'Global Production' scenario and two PV performance scenarios. The solid lines show the mean simulation result, while the bands indicate a confidence interval of 95% around the mean simulation. (**a**) Monte Carlo simulation for cumulative net energy output. (**b**) Monte Carlo simulation for cumulative net greenhouse gas emissions avoidance.

the mean simulation shows break-even before 2015. In the 'Increasing $PR$' scenario, break-even occurs before 2012 for all production location scenarios. For GHG emissions, break-even also has likely been reached already, during 2011 for the 'Increasing $PR$'/ 'Global Production' scenario combination. In the 'Low $PR$' scenario, break-even occurs during 2013. After the respective break-even points, the net energy output and emission avoidance increase rapidly. Considering we based the study on data for mono- and polycrystalline based PV system only, and the environmental footprint of other technologies is generally smaller[30], the break-even points possibly occur even sooner. We expect this effect to be small however, as the contribution of other technologies to total installed capacity has been and will likely remain limited to under 10%, and the effect on the environmental footprint of the total installed PV capacity will thus be small as well.

We analysed the effect of uncertainty in the learning curve models by means of Monte Carlo analysis. The results, plotted in Fig. 5, show that for the 'Global Production' scenario, break even in terms of energy occurs with a likelihood of 95% between 1997–2015 and between 2009–2018 for the 'Increasing $PR$' and 'Low $PR$' scenarios, respectively. For GHG emissions, the respective ranges are 2001–2016 and 2005–2019.

## Discussion

In this study we have analysed the development of the PV industry in terms of cost, cumulative energy demand, greenhouse gas emissions, and cumulative installed PV capacity. We have used this data to determine the net contribution of the PV industry in terms of CED and GHG emissions. Our analysis relies on a chain of (sometimes interdependent) data that can be difficult to accurately determine. For instance, development of cumulative installed PV capacity is not a measured quantity, but often rather estimated at country levels by performing surveys among select PV suppliers and extrapolating the resulting data. Some, but not all countries require registration of all installed PV systems and thus have more accurate data. The complete dataset is thus an aggregate of more and less accurate data.

Another main factor is the performance of PV systems over time, which is used to determine both the energy production and GHG emission avoidance of the total installed PV capacity. The performance of PV systems can be measured directly, or inferred from high-level statistics databases showing both installed PV capacity and generated electricity, such as the EIA (http://www.eia.gov/beta/international/browser/) and UN databases[28]. The former are studies that result in detailed and accurate assessment of PV performance, but of a limited subset of PV

systems. The latter can result in very unrealistic values for PV performance when the databases for PV capacity and PV electricity production are not aligned. For instance, for the year 1992 the calculated yield of PV systems from the UN database[28] in the US is over 17,000 kWh kW$_p^{-1}$, while typical annual yields are currently in the range of 1,400–1,500 kWh kW$_p^{-1}$. Furthermore, in many cases PV electricity production is estimated from installed capacity figures by means of an estimate of the specific yield of PV capacity, rather than measured from actual production. As mentioned in the Methods section, this makes it difficult to ascertain the accuracy of the values for all countries and years. To address these issues we have analysed two performance scenarios (see Methods section): a worst-case and a realistic case. In Fig. 6 we show a comparison between the different datasets for electricity production, installed capacity and inferred global average specific yield of PV capacity (kWh kW$_p^{-1}$). Focusing on electricity production (Fig. 6a) we see that although the trends look very similar for the period between 1997 and 2014, our 'Increasing PR' scenario shows somewhat higher electricity production in the last years compared to the two databases, while the 'Low PR' scenario shows lower electricity production. The higher electricity production from the 'Increasing PR' scenario is partly due to the fact that the installed capacity numbers in both the EIA and UN databases are lower, compared to the data we use in this study (shown in Fig. 1). Examining the yield inferred from the different datasources (Fig. 6c) we see that especially in the years before 2005 the EIA database but especially the UN database data results in unrealistically high average yield numbers. Furthermore it is shown that the time-range of the data is insufficient to cover the whole time horizon of our study. Taking into account the data from Fig. 6a, we argue that it is likely that the 'Increasing PR' and 'Low PR' scenarios cover a range of results that includes those that would be obtained by using one or both of the statistics databases.

For determining the net GHG emission avoidance, we need to make assumptions on what kind of electricity is used during production of PV systems, and replaced (or avoided) by produced PV electricity, and what the GHG emission factors are of both. For PV production, we assumed the use of average grid mixes of the producing countries. We have also assumed that electricity from PV replaced the average grid electricity mix. We deem this

to be a conservative estimate, as new PV capacity (and other renewable electricity sources) is more likely to replace older, fossil fuelled power plants as they are decommissioned, mainly coal fired power plants[1], and thus avoided emissions could be larger. On the other hand, considering the timing of electricity production from PV, it could also replace electricity from more flexible sources of generation than baseline coal-fired power plants, such as gas fired power plants. As these are also often fossil fuel powered, especially in the major PV markets, we assume that using the grid average is a reasonable approximation. As the share of PV and other renewables in the electricity mix increases, the average grid GHG emission factor will decrease, and as a result, the GHG emission avoidance also decreases. Finally, increasing amounts of intermittent electricity sources like PV will require either storage or back-up capacity. In the absence of sufficient storage capacity, back-up power plants, possibly fossil-fuel fired, might limit net GHG emission avoidance. Furthermore, storage options will contribute to the environmental footprint of PV electricity as their production of course requires the input of energy and materials, but they will decrease the requirement for back-up generation.

Our results show that from 1975 onwards, there are clear trends in reduction of cost, CED, and GHG emissions, concurrently with a rapid growth of installed PV capacity. While cost decreases with 20.1% for each doubling in capacity, CED decreases with 11.9–12.6% and GHG emissions with 16.5–23.6%. The rapid growth of the PV industry has resulted in the creation of a temporary net primary energy sink, as well as it being a temporary net emitter of GHG emissions. However, because of the trend of decreasing environmental footprint concurrent with the growth of CIPC, according to the 'Increasing PR' scenario, this debt was likely already repaid in 2011 for both CED and GHG emissions. For the worst case scenario, which is the 'Low PR' scenario, the 95% confidence interval shows break-even during 2017 for CED and 2018 for GHG emissions.

## Methods

**Data for PV market development and environmental footprint of PV production.** For the study conducted here we needed several different sources of data: **1)** historical data on the development of cumulative installed PV capacity (CIPC) including PV technology market shares, **2)** PV cost data for the period under

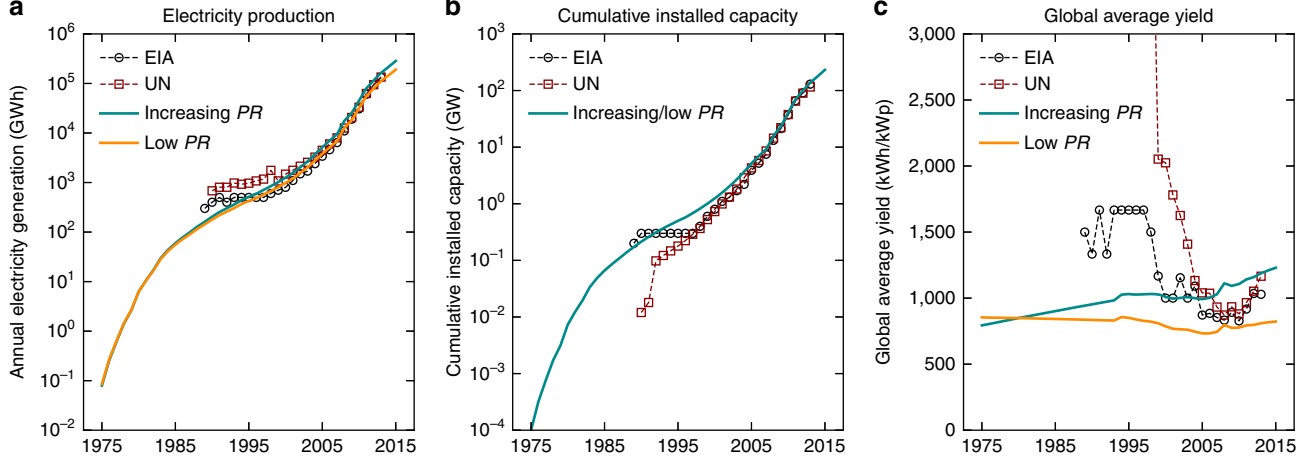

**Figure 6 | Comparison of different PV performance data sources.** (**a**) Comparison of the electricity generation as estimated by different sources of data: EIA database (http://www.eia.gov/beta/international/browser/), UN database[28] and 'Increasing PR' and 'Low PR' scenarios used in this study. (**b**) Comparison of the cumulative installed capacity from different data sources: EIA database, UN database and the data used in this study. (**c**) Comparison of the calculated global average electricity yield by using the different data sources or scenarios shown in (**a**,**b**).

investigation, **3**) forecast of the development of CIPC and PV technology market shares, **4**) life-cycle assessment (LCA) results for PV for the period studied.

Historical data for CIPC was obtained mainly from the IEA PVPS reports[4,31,32], reports from SolarPowerEurope (formerly EPIA)[1,33] and other literature data[16,25,26]. Cost data was taken from[16,25,26,34,35]; all cost data in this paper is corrected for inflation by means of the Consumer Price Index[36] and expressed in 2015 USD. Data for PV technology market shares was taken from[16]. Environmental impact data was obtained from LCA studies conducted between 1976 and 2014, shown in the Supplementary Information in Supplementary Tables 1–5. Data was filtered only to exclude 'worst-case' or 'best-case' scenarios, prospective studies, and studies that did not include results for complete PV systems (see also Supplementary Methods).

For the studies on the energy payback time and greenhouse gas footprint of PV module production, it is sometimes difficult to ascertain in retrospect whether the studies were performed using a consistent method, especially for the older studies selected here. Other meta-reviews of PV LCA's employ a stringent screening process eliminating most of the studies available[30,37]. As we are interested in development of environmental footprint over time, a similar procedure would exclude most of the studies conducted before 2000. Therefore, we have adopted a simpler screening process: the LCA studies should report CED and/or GHG emissions for a complete PV system with enough meta-information to convert the reported units to our harmonised units (see section), and should analyse existing production processes (not prospective, worst or best case processes).

**Harmonization of environmental footprint data.** In the timeframe we are analysing, the environmental footprint of PV has been studied many times. The earliest study in our analysis dates from 1976, although most studies are from after 2000. The approach with which the environmental footprint of PV was determined has been steadily improving over the years. Only in 2009, standardised methodology guidelines specifically for PV systems were published[38] as a result of an IEA PVPS project specifically focusing on the environmental impact of PV (Task 12). These guidelines were updated in 2011[21], although the practice of performing a Life Cycle Assessment (LCA) was standardised first in 1997.

For CED, we investigate megajoules of primary energy per watt-peak of PV system capacity (MJP $W_p^{-1}$). For GHG emissions, we analyse the GHG emission associated with the production of a watt-peak of PV system capacity (gCO$_2$-eq $W_p^{-1}$) but often report emissions per kWh of generated electricity as well (gCO$_2$-eq kWh$^{-1}$), as this is the unit most commonly used to express the GHG impact of PV. Where needed, conversion to the desired units was performed using harmonisation criteria based on LCA guidelines from[21]: a conversion factor from primary energy to electricity of 0.311; an insolation of 1,700 kWh·m$^{-2}$·year; a performance ratio of 0.75; a module degradation rate of 0.7% per year; and the reported system capacity (Wp). The performance ratio *PR* is defined as[39]:

$$PR = \frac{Y_f}{Y_r} \qquad (1)$$

where $Y_f$ is the final energy yield of a PV system per unit of capacity, and $Y_r$ is the reference yield per the same unit of capacity. $Y_r$ is calculated as $H_{POA}/G_{STC}$, where $H_{POA}$ is the plane of array irradiance, and $G_{STC}$ the irradiance at which PV system capacity is determined (STC conditions). Thus, the annual energy yield is given by:

$$E_{annual} = PR \times \frac{H_{POA}}{G_{STC}} \times C_{PV} \qquad (2)$$

where $C_{PV}$ is the system capacity. The ratio $H_{POA}/G_{STC}$ gives us a figure that represents equivalent annual full load hours, and thus this calculation is not dependent on the efficiency of the PV systems investigated, but only the considered system capacity. The lifetime energy yield, corrected for the assumed (linear) degradation of performance is calculated as:

$$E_{lifetime} = E_{annual} \times T_{lifetime} \times \left(1 - r_{deg} \cdot \frac{T_{lifetime}}{2}\right) \qquad (3)$$

**Experience curve.** The production costs of PV decrease with increased cumulative production, based on the theory of technological learning. The relation between cost and cumulative units of production is described by the experience curve[40]:

$$C_n = C_1 n^{-a} \qquad (4)$$

where $C_n$ is the cost of the *n*-th unit of production, $C_1$ is the cost of the first unit of production, *n* is the cumulative production volume and *a* is the 'experience parameter'[17]. The 'experience parameter' describes the decrease in cost as a function of increased cumulative production. In the context of experience curves, there is often mention of the 'learning rate', which is the cost decrease for a doubling of cumulative production. This 'learning rate' can be obtained by rewriting equation 4:

$$C_n = C_1 n^{\log_2(1-l)} \qquad (5)$$

where *l* is the 'learning rate'. The logarithm (base 2) in the exponent shows us that for each doubling of production volume, the cost of produced units decrease with a

factor *l*. In this paper, we use this relation to establish the learning rate for PV price, CED, EPBT and LCGHG emissions, by performing orthogonal distance regression analysis of the environmental impact data to this non-linear model, using the open source Python library 'SciPy' (http://www.scipy.org). Production volume for photovoltaics is most accurately reported in terms of cumulative installed PV capacity in watt-peak ($W_p$), so we will use this metric instead of cumulative number of produced units (cells, modules, systems). We have used the 'Delta method' to calculate confidence intervals for the fitted models[41].

As discussed in[17], the relationship between price and cumulative production is indirect (while that between production cost and cumulative production is direct), as market dynamics can influence the margin between cost and price. Only in a stable market phase does the price-experience curve have the same slope as a cost-experience curve[17]. However, as only price data is available for the period under study, we focus on the price-experience curve.

**Production location.** Both the production and installation location of PV influence its environmental impact. The production location mainly because the environmental impact of the electricity used in production is very locationally dependent, and as a result, production of PV in China has, for example, almost twice the GHG footprint compared to production in Europe[7,42,43]. For CED the difference is smaller but still significant. For installation, the environmental benefit of PV is larger where the environmental footprint of local electricity is greater, as it is assumed that the production of electricity from PV replaces electricity from fossil sources.

In our analysis, we have investigated the effect of production location in three scenarios: first, production in Europe; secondly, globally dispersed production, based on actual production location data from[16]; and thirdly, production in China.

To account for production location, we have combined data on the development of main PV production regions with the development of environmental impact of electricity production in those locations. For CED and GHG, we calculated a correction factor which is the average of the relative CED or GHG-footprint of electricity in each location, weighted by the share of production in each location. As the production has shifted from the US, to Europe and more recently to Asia (mainly China), this factor was calculated for each year. We also accounted for the development of the GHG-footprint of electricity over time, based on data from the UN[28] and the World Resources Institute (http://cait.wri.org). No data was found suitable to include the development of CED of electricity over time, so the relative CED for each location was assumed constant over time and was based on[44]. For production in China, the factor is based on only the relative CED and GHG-footprint of Chinese electricity (over time). For production in Europe, the factors are set to 1, as the results from the environmental impact studies are mostly based on production in Europe.

**Projections and net contribution.** From the data we have analysed we have established fitted models of development of CED, life cycle GHG emissions as a function of CIPC and their development over time. These models combined were used to calculate the total CED of PV production by integrating the learning curve. For instance, the environmental impact of production of a unit of PV in a certain year is given by:

$$EI_t(y) = \frac{\int_{y-1}^{y} EI_{1;t} \cdot C_t(y)^{\log_2(1-l_t)} \, dy}{C_t(y) - C_t(y-1)} \qquad (6)$$

where $EI_{1;t}$ is the environmental impact of the first unit of production of technology *t* (see also $C_1$ in equation 4), $C_t$ is the cumulative installed PV capacity of technology *t* in year *y*, and $l_t$ is the learning rate of that technology. For each year, the environmental impact is calculated for mono- and polycrystalline silicon-based PV systems. To calculate the total annual environmental impact from PV we extrapolate the results for mono- and polycrystalline PV to total installed capacity, and for production location:

$$EI_{released}(y) = \sum_{t,i} EI_t(y) \times {}^P C_{t,i}(y) \cdot f_{EI;t;i}(y) \qquad (7)$$

where ${}^P C_{t,i}(y)$ is the PV system capacity of technology *t* produced in country *i* in year *y*, and $f_{EI;t;i}(y)$ is a factor relating the environmental impact of production of PV in location *i* to the baseline results obtained from the learning curve, and is calculated as:

$$f_{EI;t;i}(y) = \left(\left(\frac{EI_{elec;t;i}(y)}{EI_{elec;t;base}} - 1\right) \cdot f_{elec;t;PV} + 1\right) \cdot EI_t \qquad (8)$$

where $EI_{elec;t;i}$ and $EI_{elec;t;base}$ are the environmental impact of electricity production in country *i* and the baseline scenario, respectively, and $f_{elec;t;PV}$ is the fraction of environmental impact related to electricity use in production, taken from[45]. For GHG emissions $EI_{elec}$ is calculated from databases from the UN[28] and the World Resources Institute (http://cait.wri.org), for CED historic data was not available, and we assumed a constant factor between countries based on data from the ecoinvent database[44]. Thus, we account for the effect of production location on environmental impact by varying the impact of electricity production. We assume

here that direct electricity use in the lifecycle of PV production, from silicon to PV system, is changed from the baseline to the country average.

The cumulative net environmental impact of PV is then calculated, for CED and GHG separately as:

$$EB_{net;cumulative} = \sum_{y=1975}^{y_{end}} EI_{avoided}(y) - EI_{released}(y) \qquad (9)$$

where $EI_{avoided}(y)$ is calculated based on installed capacity shares from[2,46,47] and given by:

$$EI_{avoided}(y) = PR \times \sum_i \frac{H_i}{G_{STC}} \cdot C_{active;i}(y) \cdot EI_{elec;i} \qquad (10)$$

where $PR$ is the Performance Ratio, $H_i$ is the population weighted plane-of-array insolation in country $i$, $G_{STC}$ is the standard testing condition irradiance ($1,000\,W\,m^{-2}$), and $C_{active;i}$ is the active installed PV capacity in country $i$. See also equations (1 and 2). The active capacity was calculated by correcting the cumulative installed capacity figures with an assumed degradation rate of 0.7% per year and a lifetime of 25 years.

The $PR$ is an important metric relating the actual yield of a PV system to the theoretical yield calculated with just the annual insolation and the systems' rated (peak) power. It takes into account loss factors like higher operating temperatures, inverter and cabling losses, and other losses such as due to soiling, periods of outages, and suboptimal orientation. There likely is a trend of $PR$ versus time, as increasing knowledge about and monitoring of PV performance, as well as improved system design and inverter efficiencies have led to an increase in system yields, as shown in[39,48]. Accurate data on the actual performance of CIPC is however practically non-existent. There are in general two approaches in determining the actual $PR$ of CIPC figure: a top-down analysis combining installed capacity figures with electricity production figures for PV installations, and a bottom-up, detailed analysis of PV performance using dedicated test facilities or a limited number of PV systems. Data for the former approach is readily available, but lacks in geographical and temporal scope. For instance, statistics from both the U.S. Energy Information Administration (EIA, http://www.eia.gov/beta/international/browser/) and the UN statistics database[28] are insufficient to result in timeseries, from 1975 to now, of country-specific $PR$ values, or other metrics that allow us to calculate historical energy generation from CIPC. The data goes back only to 1990, and for most countries, data is only available from 2010 onwards. Furthermore, especially for the older data, accuracy seems to be very low as unrealistic yield figures are obtained especially from the UN database (see also the Discussion section). Bottom-up studies result in very reasonable $PR$ figures (around 75–85%) but their scope is even more limited (in terms of period and geography). More recently, due to improved output monitoring of PV systems (e.g., by embedded monitoring solutions in PV inverters) the amount of available data has increased and studies are published on the performance of large numbers of privately owned PV systems[49,50]. Unfortunately, the results from these studies are also (still) very limited in geographical but especially temporal scope, due to unavailability of older data. As $PR$ is a very significant factor in determining annual yields, we have analysed two separate scenarios for the development of $PR$ over time, which we consider represent a worst-case and more-likely scenario:

● A constant low-$PR$ scenario with a $PR$ of 0.5 based on the lowest estimate mentioned in literature referring to general PV performance.

● A increasing $PR$ scenario, with $PR$ increasing linearly with respect to time from 0.5 in 1975 to a maximum of 0.8 for the years 2015 and later, for all countries.

Aside from a trend in time for PR, there is also a variation of PR per location, as ambient temperature, but also spectral variations have effect on the performance of PV systems. Statistics on the actual performance of systems in all countries analysed, for the whole period studied, are however much too limited or inaccurate. Therefore we have assumed an equal $PR$ for all locations.

Insolation data ($H_i$) was taken from[51]. We opted to use data that gives population-density weighted country insolation for surfaces with a fixed tilt and optimal orientation. We thus assume most systems to be installed in or near population centra, as is common at least in large parts of Europe. For some locations this might not be accurate. For instance, in the USA, large PV installations are built at locations in the South-West of the country, while population centres can be more confined to lower irradiance area's like the 'Boston-Washington Corridor', which has a population of almost 50 million, or 15.4% of the total US population.

**Monte Carlo analysis.** The parameters of the nonlinear models fitted to the data ($EI_t$ and $l_t$ in equation (6)) have a certain standard error as a result of deviation of the data from the fitted model, which are established by the fitting tool by determining the covariance of the fit parameters. We use Monte Carlo simulation to analyse the effect of these fit errors on our results. For each calculation, we generated 10,000 random samples of the parameters from a normal distribution for which the mean is the fitted parameter value and the standard deviation is the calculated error in the parameter. We then recalculated the results using these samples, and present the intervals that cover the range from 2.5th percentile to the 97.5th percentile of the results, e.g., a 95% confidence interval around the main result.

**Data availability.** The authors declare that all the data supporting the findings of this study are available within the article and its Supplementary Information files and from the corresponding authors upon reasonable request.

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

## Acknowledgements

This work was performed within 'FLASH', a 'Perspectief' programme funded by Technology Foundation STW (www.stw.nl/en) under project number 12172.

## Author contributions

A.L. conceptualised the study, gathered data, performed the analyses, designed the figures and wrote the paper. W.G.J.H.M.v.S guided the project. W.G.J.H.M.v.S, A.P.C.F. and R.E.I.S. offered input and concepts for A.L to write the paper.
