## [Peer Review File · Nature Communications]

Reviewer #1 (Remarks to the Author)

The paper investigates the question as to whether the global capacity of photovoltaic panels installed has reached net positive energy balance. This point is sometimes used to question the usefulness of PV electricity and in this paper it is shown that there is now a strong net positive energy generation. This is an important result which is likely to be picked up in the popular press and the authors provide much useful supporting data.

The authors acknowledge that similar studies have been performed in the past [Ref 14] but point out that this work merely predicted that the point of net positive generation is close, while the present authors show that it has now arrived.

For a publication in Nature communications, it would be useful if the authors provided some further discussion on the issue of maintaining growth rates below 1/PBT. On p2 it is simply stated that the 'industry should be limited to growth rates below 1/PBT'. While it is true that 1/PBT defines the threshold for a net positive effect, there may be circumstances where it is desirable for industry to exceed this growth rate, e.g. in order to invest manufacturing capacity early and benefit from much larger production volume later. The degree to which 1/PBT should be exceeded and for how long depends of course on many other factors that are beyond the scope of the paper, but the statement as it stands is misleading. Indeed, the very premise of this paper is that exceeding 1/PBT has a net positive effect after a couple of years when growth slows. It would be better to make the point about the importance of considering the future dynamics before passing judgment on 1/PBT. The authors may like to reference a recent study by Emmott et al that considered this point from a carbon perspective

Emmott, C.J.M., Ekins-Daukes, N.J. & Nelson, J., 2014. Dynamic carbon mitigation analysis: the role of thin-film photovoltaics. *Energy & Environmental Science*, 7(6), pp.1810-1818.

The analysis of the data appears to be reasonable and the graphs presented are well constructed. I feel the authors could usefully reference one of the original research papers on this problem

Lysen, E. & Daey Ouwens, C. (2002) Energy effects of the rapid growth of PV capacity and the urgent need to reduce the energy payback time. Paper presented at the PV in Europe, From PV Technology to Energy Solutions Conference, Rome, 7-11 October 2002 .

Reviewer #2 (Remarks to the Author)

A. Summary of the key results

The work provides updated parameter values for learning curves of PV technologies in terms of price, cumulative energy demand (CED) and greenhouse gas (GHG) emissions. This information is then used to determine the net energy and GHG payback year for the PV industry as a whole for a range of scenarios.

B. Originality and interest: if not novel, please give references

The CED portion is essentially an update of work by Dale and Benson (2013) The energy balance of the global PV industry. The GHG portion is novel.

C. Data & methodology: validity of approach, quality of data, quality of presentation

Data is as good as it could be. Approach is adequate, with some issues discussed below.

D. Appropriate use of statistics and treatment of uncertainties

Yes

E. Conclusions: robustness, validity, reliability

Lack of validation/comparison of results to historical data for PV electricity generation (e.g. from UN, EIA, IEA)

F. Suggested improvements: experiments, data for possible revision

See detailed comments below.

G. References: appropriate credit to previous work?

Some missing references to previous work in this area, e.g. Emmott, C. J. M., Ekins-Daukes, N. J., & Nelson, J. (2014). Dynamic carbon mitigation analysis: the role of thin-film photovoltaics. *Energy & Environmental Science*, 7(6), 1810-1818.

H. Clarity and context: lucidity of abstract/summary, appropriateness of abstract, introduction and conclusions

All good.

Detailed comments:

Title is a little confusing, as the data seems to cover 1975-2015, a period of 40 years.

Page 2, line 46: Energy cannot be 'consumed'. Additionally, though the net energy and net GHG issues are correlated, they are not necessarily so. There could be a situation where the PV industry was a net energy sink, yet was providing GHG benefits. Conversely, the PV industry could provide net energy and still be a positive emitter of GHGs.

Page 2, line 47: "To avoid this, growth of the industry should be limited by 1/PBT". This should be referenced to Grimmer, D. P. (1981). Solar energy breeders. *Solar Energy*, 26(1), 49-53.

Page 3, line 60: "A similar approach was used before by Dale and Benson [14], who focused on net electricity, and analysed studies in a narrower timeframe between 2000 and 2010" Dale and Benson drew on studies from 1990 to 2010.

Page 3, line 68: "Then" should be "than"

Page 3, line 71: "Still, especially for energy pay-back time (which is calculated from cumulative energy demand)". In order to calculate EPBT from CED [MJ/Wp] we need the module (or system) efficiency.

Page 3, line 87: "we see that these meta-reviews cannot account for the fast improvement of the environmental footprint of PV systems, e.g. the environmental impact has significantly dropped within the period reviewed." I find this statement a little confusing. You are comparing a study scenario based only on 'state-of-the-art' manufacturing in Europe (Wetzel) with the average value from a meta-analysis of all studies from 2005-2013.

Page 4, line 97: Since you use the term 'price-experience curve' but label the graph 'cost', as some point you should discuss the difference between these two. Another interpretation of the "strong drop in price" was that there was oversupply in the PV market that had nothing to do with cost reductions. Page 4, Fig. 2: It seems strange that the 'cost' chart has data for mono- and polySi commingled, but the other charts have the data separated. Similarly, for Fig. 3, it is odd to

have one experience curve for both technologies. It is not clear that (i) there is one 'price' for both technologies and (ii) that learning in one will bring gains in the other. Additionally, the blue hexagons from 2a come from 'This study', but I do not see them presented in the appendix.

Page 5, Fig 3: "The shaded areas show a 95% confidence interval obtained from a Monte Carlo analysis of the learning curve parameters." I am pretty certain that the 'Cost' graph shaded area does not represent a 95% confidence interval, since many of the data points lie outside of it. Additionally, this is not really how I understand a 'Monte Carlo' analysis, which is normally for prediction. I would call this a regression analysis. What do the red crosses in the other four plots represent?

Page 5, line 102: "For CED, both technologies show a downward trend of CED vs. installed capacity, with learning rates of $12.6 \pm 0.85\%$ and $11.9 \pm 1.04\%$ for poly and mono-Si systems." How does this compare with earlier estimates?

Page 5, line 103: "The quality of the fit is lower compared to that for cost, likely due to a variety of methods employed to calculate the CED". Are these differences due solely to 'methodological' differences, or do they reflect differences in the real world? Since the data for 'cost' is based on module shipping price (which has a strong pressure to converge, since the PV market is global and the prices are well advertised), what reason is there that the CED or GHG values should be similarly constrained?

Page 6, line 117: "This is likely due to the fact that mono-crystalline silicon PV module production is more energy intensive, and thus benefits more from both energy usage reduction, but also from reduction of the GHG footprint of electricity production over time, compared to production of polycrystalline PV modules. Another reason for the higher learning rate of mono-Si could be the higher relative increase in module efficiency observed for this technology compared to poly-Si." This reasoning does not follow. If the GHG emissions are declining at a faster rate than the energy it must be (i) non-energy related GHGs (likely negligible) are being reduced; and/or (ii) energy inputs are being shifted to less carbon intense forms. Page 6, line 133: "For environmental impact, cumulative energy demand drops to 11.5-13.3 MJp/Wp for poly and 14.9-17.2 MJp/Wp for mono-crystalline based PV systems, equating to energy pay-back times of 135 0.8-0.9 and 1.0-1.2 years, respectively. Greenhouse gas emissions are extrapolated to drop to 0.40- 0.49 and 0.27-0.37 kgCO₂-eq/Wp or 12-14 and 8.0-11 gCO₂-eq/kWh for poly and mono-crystalline systems, respectively." This reduction assumes that the monocrystalline energy inputs are at an energy intensity of $0.27/14.9 = 0.018$ to $0.37/17.2 = 0.022$ kgCO₂-eq/MJp. Since inputs are primarily in the form of electricity, this equates to 0.22-0.26 kgCO₂-eq/kWh. This compares with current carbon intensity of electricity in China of 1.05 kgCO₂-eq/kWh and 0.39 for EU. The question then becomes, can the electricity system decarbonize sufficiently to keep up with these predictions?

Page 7, Fig.4: It seems a little disingenuous to plot the 'net energy output' in terms of 'primary energy' when the PV systems would not actually produce this amount of energy. At the least, the word 'equivalent' should make it in there somewhere. Additionally, there should be historical data points on there for comparison.

Page 8, line 167: "For instance, development of cumulative installed PV capacity is often estimated at country levels, as not all countries require registration of all installed PV systems." I'm not entirely clear what this means.

Page 8, line 174: "The latter result in very unrealistic values for PV performance." Not clear why this is the case. Needs justification, or reference to methodology section.

Page 9, line 183: "new PV capacity (and other renewable electricity sources) are more likely to replace older, fossil fuelled power plants as they are decommissioned, mainly coal fired power plants [1], and thus avoided 185 emissions are larger." While this may be true, PV electricity doesn't offset purely on the basis of installation/decommission, but at the time-of-production. As such, PV is most likely to offset natural gas peaker plants or hydro, rather than baseline coal.

Page 10, line 235: "Where needed, conversion to the desired units was performed using harmonisation criteria based on LCA guidelines from [12]: 1) a conversion factor from primary energy to electricity of 0.311; 2) an insolation of 1700 kWh·m⁻²·year; 3) a performance ratio (PR) of 0.75, and 4) a module degradation rate of 0.7%/year" This information would not allow to convert between CED or GHG on a per Wp basis without also knowing the efficiency of the module.

Page 11, Eqn. 1: It is weird to me to show an equation with a log in the exponent. I find it hard to make sense of what that means. Is there another way to represent the curve?

Page 11, line 256: "For installation, the environmental benefit of PV becomes larger as the environmental footprint of local electricity increases". This should read, "For installation, the environmental benefit of PV is larger where the environmental footprint of local electricity is greater".

Page 11, line 257: "as it is assumed that the production of electricity from PV replaces electricity from fossil sources." As discussed earlier, I'm not sure this assumption is valid. More discussion is needed.

Page 12, line 295: What is 'population weighted insolation'? Why would PV only be installed where there is population?

Page 13, line 309: "For instance, we have combined statistics from both the U.S. Energy Information Administration (EIA)[76] and the UN statistics database [69] with the installed capacity figures from the IEA PVPS [4, 8] to calculate country- and year-specific PR figures. The values obtained ranged from under 1% to over 800%." With this approach, (i) you should only make calculations within consistent datasets (since different assumptions across datasets will have a large impact) and (ii) it is unclear how you would account for the insolation which the panels receive and thus the PR value is underdetermined. Alternatively, you could calculate the capacity factor, but would have to adjust the data for growth rates. This issue is discussed in Dale and Benson.

Page 13, line 323: "A increasing PR scenario, with PR increasing linearly [with respect to what, time?] from 0.5 in 1975 to 0.8 in 2015 and further, for all countries". This is unphysical, since at some point the PR would be greater than 1. If increasing as a function of time, then by 2040 the PR would be almost 1. I would suggest a logarithmic curve.

Page 13, lines 330 and 332: Plural of 'centre' is 'centres'.

Reviewer #3 (Remarks to the Author)

The article provides a synthesis, review, and update to learning rates, cumulative net energy, and cumulative net GHG emissions for PV.

The exercise is arguably incremental in nature, and I consider the key findings to be an update rather a major advance. As the authors recognise, Dale and Benson, have previously reported cumulative net energy albeit over a different time scale. The concept of a temporary energy sink and GHG source from the rapid growth of PV is well known and discussed in the PV community. For example, figure 4 in:

[dx.doi.org/10.1021/es502542a](https://doi.org/10.1021/es502542a) | Environ. Sci. Technol. 2014, 48, 10010–10018

Nevertheless, it is always useful to see established ideas quantified and updated.

The article is well written and would be of interest to readers from a range of fields.

Some minor corrections:

(A) In the abstract we are told of learning rates of 17 to 24%, without an indication of the meaning of the range (from the main text I understand this is poly and mono Si). Include the meaning of the range or a different metric.

(B) The abbreviations GHG and CED are defined multiple times when they only need to be defined on their first occurrence.

(C) The abbreviation PR (performance ratio) appears not to be defined at its first occurrence but in section VID. (I might have missed this).

(D) In the discussion of carbon payback (section II) the solar irradiation, and hence the location, is of critical importance. One of the other should be states.

(E) The y-axis titles and descriptions in the captions in figure 4 look inconsistent. In particular the y-axis is labelled "net GHGe avoidance" while the caption says "GHG emissions".

Reviewers' comments:

We like to sincerely thank the reviewers for their very valuable comments. We have addressed them to the best of our power, see below. The changes have been marked in yellow in the revised manuscript.

Reviewer #1 (Remarks to the Author):

The paper investigates the question as to whether the global capacity of photovoltaic panels installed has reached net positive energy balance. This point is sometimes used to question the usefulness of PV electricity and in this paper it is shown that there is now a strong net positive energy generation. This is an important result which is likely to be picked up in the popular press and the authors provide much useful supporting data.

The authors acknowledge that similar studies have been performed in the past [Ref 14] but point out that this work merely predicted that the point of net positive generation is close, while the present authors show that it has now arrived.

For a publication in Nature communications, it would be useful if the authors provided some further discussion on the issue of maintaining growth rates below 1/PBT. On p2 it is simply stated that the 'industry should be limited to growth rates below 1/PBT'. While it is true that 1/PBT defines the threshold for a net positive effect, there may be circumstances where it is desirable for industry to exceed this growth rate, e.g. in order to invest manufacturing capacity early and benefit from much larger production volume later. The degree to which 1/PBT should be exceeded and for how long depends of course on many other factors that are beyond the scope of the paper, but the statement as it stands is misleading. Indeed, the very premise of this paper is that exceeding 1/PBT has a net positive effect after a couple of years when growth slows. It would be better to make the point about the importance of considering the future dynamics before passing judgment on 1/PBT. The authors may like to reference a recent study by Emmott et al that considered this point from a carbon perspective

The text has been updated to reflect the importance of considering growth dynamics when talking about 1/PBT

Emmott, C.J.M., Ekins-Daukes, N.J. & Nelson, J., 2014. Dynamic carbon mitigation analysis: the role of thin-film photovoltaics. Energy & Environmental Science, 7(6), pp.1810-1818.

The analysis of the data appears to be reasonable and the graphs presented are well constructed. I feel the authors could usefully reference one of the original research papers on this problem

Lysen, E. & Daey Ouwens, C. (2002) Energy effects of the rapid growth of PV capacity and the urgent need to reduce the energy payback time. Paper presented at the PV in Europe, From PV Technology to Energy Solutions Conference, Rome, 7-11 October 2002 .

The paper has been updated to include reference to the mentioned papers for several arguments

Reviewer #2 (Remarks to the Author):

A. Summary of the key results

The work provides updated parameter values for learning curves of PV technologies in terms of price, cumulative energy demand (CED) and greenhouse gas (GHG) emissions. This information is then used to determine the net energy and GHG payback year for the PV industry as a whole for a range of scenarios.

B. Originality and interest: if not novel, please give references

The CED portion is essentially an update of work by Dale and Benson (2013) *The energy balance of the global PV industry*. The GHG portion is novel.

C. Data & methodology: validity of approach, quality of data, quality of presentation

Data is as good as it could be. Approach is adequate, with some issues discussed below.

D. Appropriate use of statistics and treatment of uncertainties

Yes

E. Conclusions: robustness, validity, reliability

Lack of validation/comparison of results to historical data for PV electricity generation (e.g. from UN, EIA, IEA)

F. Suggested improvements: experiments, data for possible revision

See detailed comments below.

G. References: appropriate credit to previous work?

Some missing references to previous work in this area, e.g. Emmott, C. J. M., Ekins-Daukes, N. J., & Nelson, J. (2014). *Dynamic carbon mitigation analysis: the role of thin-film photovoltaics*. *Energy & Environmental Science*, 7(6), 1810-1818.

H. Clarity and context: lucidity of abstract/summary, appropriateness of abstract, introduction and conclusions

All good.

Detailed comments from attached file:

Review of Louwen et al (2016) *50 Years of PV Development: Review, Learning Rates and Outlook for Cost and Environmental Footprint*

Title is a little confusing, as the data seems to cover 1975-2015, a period of 40 years.

(Some of) the data originally spanned a longer period, but the reviewer is correct that currently it does not. We have updated the title accordingly.

Page 2, line 46: Energy cannot be 'consumed'. Additionally, though the net energy and net GHG issues are correlated, they are not necessarily so. There could be a situation where the PV industry was a net energy sink, yet was providing GHG benefits. Conversely, the PV industry could provide net energy and still be a positive emitter of GHGs.

The text has been updated to show 'energy embedded in PV' rather than 'consumed'. Furthermore, CED and GHG have been mentioned separately, including a remark that they can be correlated, but are not necessarily so.

Page 2, line 47: "To avoid this, growth of the industry should be limited by 1/PBT". This should be referenced to Grimmer, D. P. (1981). *Solar energy breeders*. *Solar Energy*, 26(1), 49-53.

The reference has been added to the text

Page 3, line 60: "A similar approach was used before by Dale and Benson [14], who focused on net electricity, and analysed studies in a narrower timeframe between 2000 and 2010" Dale and Benson drew on studies from 1990 to 2010.

The text has been corrected accordingly

Page 3, line 68: "Then" should be "than"
The text has been corrected accordingly

Page 3, line 71: "Still, especially for energy pay-back time (which is calculated from cumulative energy demand)". In order to calculate EPBT from CED [MJ/Wp] we need the module (or system) efficiency. In this case we have calculated EPBT from system CED by dividing the CED/Wp for a complete system by the primary energy equivalent production of 1 Wp of PV system, based on an irradiance of 1700 kWh/m² and a performance ratio of 0.75.

Page 3, line 87: "we see that these meta-reviews cannot account for the fast improvement of the environmental footprint of PV systems, e.g. the environmental impact has significantly dropped within the period reviewed." I find this statement a little confusing. You are comparing a study scenario based only on 'state-of-the-art' manufacturing in Europe (Wetzel) with the average value from a meta-analysis of all studies from 2005-2013.

This sentence was meant to indicate that the developments in PV environmental impact are so fast that the results from these meta-studies differ very substantially from state-of-the art. The text has been rewritten to more accurately reflect this.

Page 4, line 97: Since you use the term 'price-experience curve' but label the graph 'cost', as some point you should discuss the difference between these two. Another interpretation of the "strong drop in price" was that there was oversupply in the PV market that had nothing to do with cost reductions.

As we only use price data, the figure has been changed accordingly (now states "cost"). Furthermore, we have added a short remark in the methods section VIB about the difference between cost and price-experience.

Page 4, Fig. 2: It seems strange that the 'cost' chart has data for mono- and poly- Si commingled, but the other charts have the data separated. Similarly, for Fig. 3, it is odd to have one experience curve for both technologies. It is not clear that (i) there is one 'price' for both technologies and (ii) that learning in one will bring gains in the other. Additionally, the blue hexagons from 2a come from 'This study', but I do not see them presented in the appendix.

The blue hexagons represent data from two sources. Captions and graphs have been updated to accurately show these sources, instead of just "this study". The "cost" (now "price") graph has data combined as we did not have any data available that separates these two, while for the LCA we did have this data separate.

Page 5, Fig 3: "The shaded areas show a 95% confidence interval obtained from a Monte Carlo analysis of the learning curve parameters." I am pretty certain that the 'Cost' graph shaded area does not represent a 95% confidence interval, since many of the data points lie outside of it. Additionally, this is not really how I understand a 'Monte Carlo' analysis, which is normally for prediction. I would call this a regression analysis. What do the red crosses in the other four plots represent?

The determination of a confidence interval for a nonlinear regression is not as straightforward as it is for a linear regression. Therefore, we used Monte Carlo analysis to estimate a confidence interval. In the revised paper we have changed this approach and calculated the confidence interval using the delta method. This is based on partial derivate vectors of the fitted model to the estimated model parameters and the covariance matrix of the fitted model. The methods include a mention of this approach with a reference to this method. The red crosses showed LCA results from prospective studies. These are now omitted as they weren't mentioned and did not contribute to the discussion in the paper.

Page 5, line 102: "For CED, both technologies show a downward trend of CED vs. installed capacity, with learning rates of 12.6±0.85% and 11.9±1.04% for poly and mono-Si systems." How does this compare with earlier estimates?

Comparison with a previous study has been added to the text.

Page 5, line 103: "The quality of the fit is lower compared to that for cost, likely due to a variety of methods employed to calculate the CED". Are these differences due solely to 'methodological' differences, or do they reflect differences in the real world? Since the data for 'cost' is based on module shipping price (which has a strong pressure to converge, since the PV market is global and the prices are well advertised), what reason is there that the CED or GHG values should be similarly constrained?

Indeed it is true that the CED data likely reflects actual differences in the real world, as they often are not general/average numbers, but rather case studies on a specific production process and resulting CED. So indeed this data would not converge like the price data. The text has been updated to present this argument.

Page 6, line 117: *“This is likely due to the fact that mono-crystalline silicon PV module production is more energy intensive, and thus benefits more from both energy usage reduction, but also from reduction of the GHG footprint of electricity production over time, compared to production of polycrystalline PV modules. Another reason for the higher learning rate of mono-Si could be the higher relative increase in module efficiency observed for this technology compared to poly-Si.” This reasoning does not follow. If the GHG emissions are declining at a faster rate than the energy it must be (i) non-energy related GHGs (likely negligible) are being reduced; and/or (ii) energy inputs are being shifted to less carbon intense forms.*

Given the higher fraction of input electricity in the production of monocrystalline silicon, we feel an external reduction of the carbon intensity of input energy will have more effect on the GHG footprint of mono compared to polycrystalline silicon. We think we agree with the reviewer on this point, but our wording was poorly chosen and thus did not accurately reflect this. The text has been updated to improve the wording.

Page 6, line 133: *“For environmental impact, cumulative energy demand drops to 11.5-13.3 MJp/Wp for poly and 14.9-17.2 MJp/Wp for mono-crystalline based PV systems, equating to energy pay-back times of 135 0.8-0.9 and 1.0-1.2 years, respectively. Greenhouse gas emissions are extrapolated to drop to 0.40- 0.49 and 0.27-0.37 kgCO₂-eq/Wp or 12-14 and 8.0-11 gCO₂-eq/kWh for poly and mono-crystalline systems, respectively.” This reduction assumes that the mono- crystalline energy inputs are at an energy intensity of $0.27/14.9 = 0.018$ to $0.37/17.2 = 0.022$ kgCO₂-eq/MJp. Since inputs are primarily in the form of electricity, this equates to 0.22-0.26 kgCO₂-eq/kWh. This compares with current carbon intensity of electricity in China of 1.05 kgCO₂-eq/kWh and 0.39 for EU. The question then becomes, can the electricity system decarbonize sufficiently to keep up with these predictions?*

This reasoning indeed implies that future PV production at the GHG intensity projected by the learning curves requires a strong decrease of the GHG intensity of input energy. Targets that are put in place by governments should result in sufficiently strong decreases of GHG intensity of electricity. A short remark has been added.

Page 7, Fig.4: *It seems a little disingenuous to plot the ‘net energy output’ in terms of ‘primary energy’ when the PV systems would not actually produce this amount of energy. At the least, the word ‘equivalent’ should make it in there somewhere. Additionally, there should be historical data points on there for comparison.*

The figures (4 and 5) and their captions have been updated to reflect that indeed we talk about primary energy equivalent energy, not actual energy produced. As these graphs show simulations / model calculations, it is not possible to include historical datapoints in there, as the data plotted is a combination of historical data on installed capacity, modelled energy impact of production, scenario assumptions on the production of electricity from PV and PV degradation.

Page 8, line 167: *“For instance, development of cumulative installed PV capacity is often estimated at country levels, as not all countries require registration of all installed PV systems.” I’m not entirely clear what this means.*

The total figure for cumulative installed capacity is a combination of data from countries that register all PV systems (accurate data) and from countries that do not require registration, but for instance perform surveys among key suppliers of PV and extrapolate (as an example, for the Netherlands this is the case, resulting in quite inaccurate figures). The text has been updated to more clearly show our argument.

Page 8, line 174: *“The latter result in very unrealistic values for PV performance.” Not clear why this is the case. Needs justification, or reference to methodology section.*

Wording has been altered / expanded to more accurately reflect our rationale. Sometimes the databases cover one or the other (capacity or production), and combining two unaligned databases to calculated yield results in these cases in very high or low figures. Also, the top-down databases often estimate electricity production, based on the (estimate of) installed PV capacity and an (estimated) specific yield figure.

Page 9, line 183: *“new PV capacity (and other renewable electricity sources) are more likely to replace older, fossil fuelled power plants as they are decommissioned, mainly coal fired power plants [1], and thus avoided 185 emissions are larger.” While this may be true, PV electricity doesn’t offset purely on the basis of*

installation/decommission, but at the time-of-production. As such, PV is most likely to offset natural gas peaker plants or hydro, rather than baseline coal.

We agree with the reviewer. The text has been updated to reflect these considerations.

Page 10, line 235: "Where needed, conversion to the desired units was performed using harmonisation criteria based on LCA guidelines from [12]: 1) a conversion factor from primary energy to electricity of 0.311; 2) an insolation of 1700 kWh·m⁻²·year; 3) a performance ratio (PR) of 0.75, and 4) a module degradation rate of 0.7%/year" This information would not allow to convert between CED or GHG on a per Wp basis without also knowing the efficiency of the module.

The text has been updated

Page 11, Eqn. 1: It is weird to me to show an equation with a log in the exponent. I find it hard to make sense of what that means. Is there another way to represent the curve?

The log (with base 2) is in the exponent so that the learning rate l represents the decrease in unit cost when cumulative produced n doubles. This is often used as a more convenient way of representing cost decrease as a function of increased cumulative production volume. We have updated the text to also include the experience curve equation without the log, and a short discussion of why this curve is often represented with the log (base 2) in the exponent.

Page 11, line 256: "For installation, the environmental benefit of PV becomes larger as the environmental footprint of local electricity increases". This should read, "For installation, the environmental benefit of PV is larger where the environmental footprint of local electricity is greater".

We have updated the text accordingly

Page 11, line 257: "as it is assumed that the production of electricity from PV replaces electricity from fossil sources." As discussed earlier, I'm not sure this assumption is valid. More discussion is needed.

The discussion has been added in the paper, at the place of the previous comment on this topic.

Page 12, line 295: What is 'population weighted insolation'? Why would PV only be installed where there is population?

In our experience, for the current market, it is the case that most PV capacity is installed in or near urban areas. So, rather than taking a "area-weighted" insolation (e.g. average over whole country area), we have chosen to take insolation weighted by population density. As an extreme example, Sweden, Norway and Finland have a high population density in the southern parts of the countries at highest insolation but all span a large range of latitude, and thus the area-weighted insolation is much lower. For some countries it might be vice versa.

Page 13, line 309: "For instance, we have combined statistics from both the U.S. Energy Information Administration (EIA)[76] and the UN statistics database [69] with the installed capacity figures from the IEA PVPS [4, 8] to calculate country- and year-specific PR figures. The values obtained ranged from under 1% to over 800%." With this approach, (i) you should only make calculations within consistent datasets (since different assumptions across datasets will have a large impact) and (ii) it is unclear how you would account for the insolation which the panels receive and thus the PR value is underdetermined. Alternatively, you could calculate the capacity factor, but would have to adjust the data for growth rates. This issue is discussed in Dale and Benson.

Based on your suggestion we have looked deeper into the approach of Dale and Benson. The problem is indeed that we were using databases that are not aligned, thus resulting in very unrealistic values. If we take for instance all data (capacity and generation) from EIA we can calculate much more reasonable figures. The UN databases, combining generation and installed capacity still result in very unrealistic figures. For both, the time period of the data is not sufficient to cover our whole period of investigation. Therefore, we opted to use our two performance ratio scenarios instead. The irradiation figures we use to calculate yield from PR are for fixed tilt, optimally oriented surfaces.

Page 13, line 323: "A increasing PR scenario, with PR increasing linearly [with respect to what, time?] from 0.5 in 1975 to 0.8 in 2015 and further, for all countries". This is unphysical, since at some point the PR would be greater than 1. If increasing as a function of time, then by 2040 the PR would be almost 1. I would suggest a logarithmic curve.

The increase was assumed to be linear with respect to time, and reaches the maximum value of 0.8 (thus from 2015 onward it stays 0.8).

Page 13, lines 330 and 332: Plural of 'centre' is 'centres'.

The text has been updated accordingly

Reviewer #3 (Remarks to the Author):

The article provides a synthesis, review, and update to learning rates, cumulative net energy, and cumulative net GHG emissions for PV.

The exercise is arguably incremental in nature, and I consider the key findings to be an update rather a major advance. As the authors recognise, Dale and Benson, have previously reported cumulative net energy albeit over a different time scale. The concept of a temporary energy sink and GHG source from the rapid growth of PV is well known and discussed in the PV community. For example, figure 4 in:

[dx.doi.org/10.1021/es502542a](https://doi.org/10.1021/es502542a) | Environ. Sci. Technol. 2014, 48, 10010–10018

Nevertheless, it is always useful to see established ideas quantified and updated.

The article is well written and would be of interest to readers from a range of fields.

Some minor corrections:

(A) In the abstract we are told of learning rates of 17 to 24%, without an indication of the meaning of the range (from the main text I understand this is poly and mono Si). Include the meaning of the range or a different metric.

The text has been updated to more clearly make distinction between poly and mono and CED and GHG

(B) The abbreviations GHG and CED are defined multiple times when they only need to be defined on their first occurrence.

The text has been updated to only define the abbreviations once

(C) The abbreviation PR (performance ratio) appears not to be defined at its first occurrence but in section VID. (I might have missed this).

The abbreviation is shown at the first mention of performance ratio, which is in section II. Other occurrences have been updated to show it is not defined multiple times.

(D) In the discussion of carbon payback (section II) the solar irradiation, and hence the location, is of critical importance. One of the other should be states.

The paragraph has been updated to include mention of the dependence on irradiation / location.

(E) The y-axis titles and descriptions in the captions in figure 4 look inconsistent. In particular the y-axis is labelled "net GHGe avoidance" while the caption says "GHG emissions".

The figure has been updated accordingly

Reviewer #1 (Remarks to the Author)

This paper reports on an important topic that is frequently misunderstood and misrepresented. In my view the authors have performed a careful and considered analysis of the problem and present their results clearly. My technical concerns have been addressed in this revised paper which I consider worthy of publication in Nature Communications.

If the authors are given the opportunity to perform one further revision, I feel they could usefully comment on a recent paper "Energy Return on Energy Invested (ERoEI) for photovoltaic solar systems in regions of moderate insolation" by Ferroni and Hopkirk, Energy Policy 94 (2016) 336-344 who come to very different conclusions, mainly on account of drawing different system boundaries and some peculiar assumptions on energy intensity of labour costs. It would strengthen the impact of the author's paper if they could compare their results against this study.

Reviewer #2 (Remarks to the Author)

Manuscript is much improved and authors have addressed most comments, however there are still some issues that need to be addressed in the attached file.

Comments on response to reviewers:

Page 3, line 71: "Still, especially for energy pay-back time (which is calculated from cumulative energy demand)". In order to calculate EPBT from CED [MJ/Wp] we need the module (or system) efficiency.

In this case we have calculated EPBT from system CED by dividing the CED/Wp for a complete system by the primary energy equivalent production of 1 Wp of PV system, based on an irradiance of 1700 kWh/m² and a performance ratio of 0.75.

Unfortunately, this does not solve the issue. The CED/Wp does not tell us how much area the panel covers. In order to know this, you need the efficiency at standard irradiance, 1000 W/m². Secondly, the irradiance is incoming solar energy, not how much is converted into electricity for which again you need the panel efficiency.

When using the CED/Wp to obtain electricity output, you MUST have either the capital factor or the panel efficiency. There is no way around this.

Page 7, Fig.4: It seems a little disingenuous to plot the 'net energy output' in terms of 'primary energy' when the PV systems would not actually produce this amount of energy. At the least, the word 'equivalent' should make it in there somewhere. Additionally, there should be historical data points on there for comparison.

The figures (4 and 5) and their captions have been updated to reflect that indeed we talk about primary energy equivalent energy, not actual energy produced. As these graphs show simulations / model calculations, it is not possible to include historical datapoints in there, as the data plotted is a combination of historical data on installed capacity, modelled energy impact of production, scenario assumptions on the production of electricity from PV and PV degradation.

It's not clear to me why you cannot add historical data to these plots. I really think it would benefit the reader to be able to gauge how well the model reflects what actually happened.

Page 8, line 174: "The latter result in very unrealistic values for PV performance." Not clear why this is the case. Needs justification, or reference to methodology section.

Wording has been altered / expanded to more accurately reflect our rationale. Sometimes the databases cover one or the other (capacity or production), and combining two unaligned databases to calculated yield results in these cases in very high or low figures. Also, the top-down databases often estimate electricity production, based on the (estimate of) installed PV capacity and an (estimated) specific yield figure.

Your justification for not using 'actual' data seems weak to me. You are throwing out an "aggregate of more and less accurate data." for a wholly modeled approach. Why would this be more realistic? You need to at least present the results with the two methods.

Page 9, line 183: "new PV capacity (and other renewable electricity sources) are more likely to replace older, fossil fuelled power plants as they are decommissioned, mainly coal fired power plants [1], and thus avoided emissions are larger." While this may be true, PV electricity doesn't offset purely on the basis of installation/decommission, but at the time-of-production. As such, PV is most likely to offset natural gas peaker plants or hydro, rather than baseline coal.

We agree with the reviewer. The text has been updated to reflect these considerations.

Added text, "However, considering the timing of electricity production from PV, it will likely replace electricity from more sources of generation than baseline power" contradicts the previous sentence. You can't have your cake and eat it too!

Page 10, line 235: "Where needed, conversion to the desired units was performed using harmonisation criteria based on LCA guidelines from [12]: 1) a conversion factor from primary energy to electricity of 0.311; 2) an insolation of 1700 kWh·m⁻²·year; 3) a performance ratio (PR) of 0.75, and 4) a module degradation rate of 0.7%/year" This information would not allow to convert between CED or GHG on a per Wp basis without also knowing the efficiency of the module.

The text has been updated.

I don't see any text added. See earlier comment regarding converting between capacity (Wp) and electricity production (kWh)

Page 12, line 295: What is 'population weighted insolation'? Why would PV only be installed where there is population?

In our experience, for the current market, it is the case that most PV capacity is installed in or near urban areas. So, rather than taking a “area-weighted” insolation (e.g. average over whole country area), we have chosen to take insolation weighted by population density. As an extreme example, Sweden, Norway and Finland have a high population density in the southern parts of the countries at highest insolation but all span a large range of latitude, and thus the area-weighted insolation is much lower. For some countries it might be vice versa.

OK, I guess this makes sense (depending on your definition of ‘near’) and may make a difference for countries with a large population distribution across their north-south axis. However, I still disagree that historical data could not be used. See earlier comment.

Page 13, line 309: “For instance, we have combined statistics from both the U.S. Energy Information Administration (EIA)[76] and the UN statistics database [69] with the installed capacity figures from the IEA PVPS [4, 8] to calculate country and year-specific PR figures. The values obtained ranged from under 1% to over 800%.” With this approach, (i) you should only make calculations within consistent datasets (since different assumptions across datasets will have a large impact) and (ii) it is unclear how you would account for the insolation which the panels receive and thus the PR value is underdetermined. Alternatively, you could calculate the capacity factor, but would have to adjust the data for growth rates. This issue is discussed in Dale and Benson. Based on your suggestion we have looked deeper into the approach of Dale and Benson.

The problem is indeed that we were using databases that are not aligned, thus resulting in very unrealistic values. If we take for instance all data (capacity and generation) from EIA we can calculate much more reasonable figures. The UN databases, combining generation and installed capacity still result in very unrealistic figures. For both, the our two performance ratio scenarios instead. The irradiation figures we use to calculate yield from PR are for fixed tilt, optimally oriented surfaces.

Yes, the UN database has some garbage in there. I would like to see some discussion of comparison between the two results

Reviewer #3 (Remarks to the Author)

The authors now recognise some previous work looking at cumulative net GHG and cumulative radiative forcing (CRF), by Emmott and Ravikumar (albeit not the reference I suggested: Environ. Sci. Technol. 2014, 48, 10010–10018). They also separate their work from these articles by focussing on global industry growth over longer time periods, and hence satisfy my main concern regarding novelty. I now recommend publication.

RESPONSE TO REFEREES

40 Years of PV: Review, Learning rates and outlook for cost and environmental impact

To all reviewers and the editor: the title of the paper has been adapted to conform with the Nature Communications manuscript requirements

(title changed to: Net energy production and greenhouse gas emissions avoidance after 40 years of PV development)

Submitted to Nature Communications

Second Revision – Reviewer comments & Author responses

This document also contains some comments from the first revision round. The text is coloured according to the following system:

Reviewer comments from the second revision round

Author responses to comment from the second revision round

Comments by reviewers for the first revision round (including some quoted text from the manuscript)

Reviewer comments from the first revision round

Reviewer #1 (Remarks to the Author):

This paper reports on an important topic that is frequently misunderstood and misrepresented. In my view the authors have performed a careful and considered analysis of the problem and present their results clearly. My technical concerns have been addressed in this revised paper which I consider worthy of publication in Nature Communications.

If the authors are given the opportunity to perform one further revision, I feel they could usefully comment on a recent paper "Energy Return on Energy Invested (ERoEI) for photovoltaic solar systems in regions of moderate insolation" by Ferroni and Hopkirk, Energy Policy 94 (2016) 336-344 who come to very different conclusions, mainly on account of drawing different system boundaries and some peculiar assumptions on energy intensity of labour costs. It would strengthen the impact of the author's paper if they could compare their results against this study.

The authors would like to thank the reviewer for his/her insightful comments in both the revision rounds. The authors are aware of the recent publication by Ferroni and Hopkirk [redacted]. We will include some brief comments on the paper by Ferroni and Hopkirk in our paper here as well [redacted].

Reviewer #2 (Remarks to the Author):

Manuscript is much improved and authors have addressed most comments, however there are still some issues that need to be addressed in the attached file.

Page 3, line 71: "Still, especially for energy pay-back time (which is calculated from cumulative energy demand)". In order to calculate EPBT from CED [MJ/Wp] we need the module (or system) efficiency.

In this case we have calculated EPBT from system CED by dividing the CED/Wp for a complete system by the primary energy equivalent production of 1 Wp of PV system, based on an irradiance of 1700 kWh/m² and a performance ratio of 0.75.

Unfortunately, this does not solve the issue. The CED/Wp does not tell us how much area the panel covers. In order to know this, you need the efficiency at standard irradiance, 1000 W/m². Secondly, the irradiance is incoming solar energy, not how much is converted into electricity for which again you need the panel efficiency. When using the CED/Wp to obtain electricity output, you MUST have either the capital factor or the panel efficiency. There is no way around this.

To the authors' knowledge it is common practice to calculate the energy yield of a PV system by taking the system capacity, and multiplying it with the yearly insolation figure and the PR to account for all yield losses. See for instance Reich et al., 2012 (doi: 10.1002/pip.1219, <http://onlinelibrary.wiley.com/doi/10.1002/pip.1219/abstract>). Here it is implicit that a yearly insolation of 1700 kWh/m² results in an equivalent of 1700 full-load hours, as the rated capacity (W_p) is established at 1000 W/m². The full calculation thus becomes:

$$\text{Yield} = \text{Capacity} * (\text{insolation} / \text{STC-irradiance}) * \text{PR} = 1 W_p * (1700 \text{ kWh/m}^2 / 1 \text{ kW/m}^2) * 0.75.$$

The methods section has been adapted to more clearly state this (eqn 1-3, lines 272-280).

Page 7, Fig.4: It seems a little disingenuous to plot the 'net energy output' in terms of 'primary energy' when the PV systems would not actually produce this amount of energy. At the least, the word 'equivalent' should make it in there somewhere. Additionally, there should be historical data points on there for comparison.

The figures (4 and 5) and their captions have been updated to reflect that indeed we talk about primary energy equivalent energy, not actual energy produced. As these graphs show simulations / model calculations, it is not possible to include historical datapoints in there, as the data plotted is a combination of historical data on installed capacity, modelled energy impact of production, scenario assumptions on the production of electricity from PV and PV degradation.

It's not clear to me why you cannot add historical data to these plots. I really think it would benefit the reader to be able to gauge how well the model reflects what actually happened.

We feel we cannot present historical data points in these figures, as there are no available historical datapoints for the data presented in Figures 4 and 5, e.g. net (primary-equivalent) energy production and especially net GHGe avoidance. Although there is some data (e.g. from UN/EIA databases) on electricity production from PV, there is no historical data on the energy consumption of the PV industry, as this was measured in this case by using LCA studies, combining these in experience curves and calculating energy use from this and the growth of the PV industry. Furthermore, the data from the UN database is limited in time horizon, for most countries data is only available from 2010 onwards. Furthermore the data is extremely inaccurate especially for older years.

Page 8, line 174: "The latter result in very unrealistic values for PV performance." Not clear why this is the case. Needs justification, or reference to methodology section.

Wording has been altered / expanded to more accurately reflect our rationale. Sometimes the databases cover one or the other (capacity or production), and combining two unaligned databases to calculate yield results in these cases in very high or low figures. Also, the top-down databases often estimate electricity production, based on the (estimate of) installed PV capacity and an (estimated)

specific yield figure.

Your justification for not using 'actual' data seems weak to me. You are throwing out an "aggregate of more and less accurate data." for a wholly modeled approach. Why would this be more realistic? You need to at least present the results with the two methods.

We wanted to capture historical developments, including the historical development of performance of PV. The data from for instance the UN databases is of insufficient time horizon (e.g. going back only to 1990 for a handful of countries, and for most countries data is only available from 2010 onwards). Dale and Benson for instance used a single figure to determine energy yield, which did not vary over time like we try to do here. We have added some text to explain (lines 214-216, 375-378).

Page 9, line 183: "new PV capacity (and other renewable electricity sources) are more likely to replace older, fossil fuelled power plants as they are decommissioned, mainly coal fired power plants [1], and thus avoided emissions are larger." While this may be true, PV electricity doesn't offset purely on the basis of installation/decommission, but at the time-of-production. As such, PV is most likely to offset natural gas peaker plants or hydro, rather than baseline coal.

We agree with the reviewer. The text has been updated to reflect these considerations.

Added text, "However, considering the timing of electricity production from PV, it will likely replace electricity from more sources of generation than baseline power" contradicts the previous sentence. You can't have your cake and eat it too!

Agreed. We have changed the wording and added a sentence to show we feel that using the grid average is a reasonable approximation.

Page 10, line 235: "Where needed, conversion to the desired units was performed using harmonisation criteria based on LCA guidelines from [12]: 1) a conversion factor from primary energy to electricity of 0.311; 2) an insolation of 1700 kWh·m⁻²·year; 3) a performance ratio (PR) of 0.75, and 4) a module degradation rate of 0.7%/year" This information would not allow to convert between CED or GHG on a per Wp basis without also knowing the efficiency of the module.

The text has been updated

I don't see any text added. See earlier comment regarding converting between capacity (Wp) and electricity production (kWh)

Please see also our answer to an earlier remark. The methods section has been adapted to more clearly state our approach (eqn 1-3, lines 272-280).

Page 12, line 295: What is 'population weighted insolation'? Why would PV only be installed where there is population? In our experience, for the current market, it is the case that most PV capacity is installed in or near urban areas. So, rather than taking a "area-weighted" insolation (e.g. average over whole country area), we have chosen to take insolation weighted by population density. As an extreme example, Sweden, Norway and Finland have a high population density in the southern parts of the countries at highest insolation but all span a large range of latitude, and thus the area-weighted insolation is much lower. For some countries it might be vice versa.

OK, I guess this makes sense (depending on your definition of 'near') and may make a difference for countries with a large population distribution across their north-south axis. However, I still disagree that historical data could not be used. See earlier comment.

Please see also our response to a previous remark. We have screened the available historical data and deemed it insufficient for our calculations here.

Page 13, line 309: "For instance, we have combined statistics from both the U.S. Energy Information Administration (EIA)[76] and the UN statistics database [69] with the installed capacity figures from the IEA PVPS [4, 8] to calculate country- and year-specific PR figures. The values obtained ranged from under 1% to over 800%." With this approach, (i) you should only make calculations within consistent datasets (since different assumptions across datasets will have a large impact) and (ii) it is unclear how you would account for the insolation which the panels receive and thus the PR value is underdetermined. Alternatively, you could calculate the capacity factor, but would have to adjust the data for growth rates. This issue is discussed in Dale and Benson.

Based on your suggestion we have looked deeper into the approach of Dale and Benson. The problem is indeed that we were using databases that are not aligned, thus resulting in very unrealistic values. If we take for instance all data (capacity and generation) from EIA we can calculate much more reasonable figures. The UN databases, combining generation and installed capacity still result in very unrealistic figures. For both, the our two performance ratio scenarios instead. The irradiation figures we use to calculate yield from PR are for fixed tilt, optimally oriented surfaces.

Yes, the UN database has some garbage in there. I would like to see some discussion of comparison between the two results

Please see also our response to a previous remark. We have screened the available historical data and deemed it insufficient for our calculations here.

Reviewer #3 (Remarks to the Author):

The authors now recognise some previous work looking at cumulative net GHG and cumulative radiative forcing (CRF), by Emmott and Ravikumar (albeit not the reference I suggested: Environ. Sci. Technol. 2014, 48, 10010–10018). They also separate their work from these articles by focussing on global industry growth over longer time periods, and hence satisfy my main concern regarding novelty. I now recommend publication.

The reference by Ravikumar was erroneously the wrong one (problem with reference manager). We now cite the suggested reference. The authors would like to sincerely thank the reviewer for his/her insightful comments and very much appreciate the comments here and the recommendation for publication in Nature Communications. We have updated our article to include reference to the article the reviewer suggested in the first revision

[Editorial note: the editor requested specific information from the authors, in discussion with Referee #2]

RESPONSE TO REFEREES

40 Years of PV: Review, Learning rates and outlook for cost and environmental impact

To all reviewers and the editor: the title of the paper has been adapted to conform with the Nature Communications manuscript requirements

(title changed to: Net energy production and greenhouse gas emissions avoidance after 40 years of PV development)

Submitted to Nature Communications

Second Revision – Reviewer comments & Author responses

This document also contains some comments from the first revision round. The text is coloured according to the following system:

Review comments received from the editor on Sept 21st.

Author responses to comments received from the editor on Sept 21st

Reviewer comments from the second revision round

Author responses to comment from the second revision round

Comments by reviewers for the first revision round (including some quoted text from the manuscript)

Reviewer comments from the first revision round

Reviewer #1 (Remarks to the Author):

This paper reports on an important topic that is frequently misunderstood and misrepresented. In my view the authors have performed a careful and considered analysis of the problem and present their results clearly. My technical concerns have been addressed in this revised paper which I consider worthy of publication in Nature Communications.

If the authors are given the opportunity to perform one further revision, I feel they could usefully comment on a recent paper "Energy Return on Energy Invested (ERoEI) for photovoltaic solar systems in regions of moderate insolation" by Ferroni and Hopkirk, Energy Policy 94 (2016) 336-344 who come to very different conclusions, mainly on account of drawing different system boundaries and some peculiar assumptions on energy intensity of labour costs. It would strengthen the impact of the author's paper if they could compare their results against this study.

The authors would like to thank the reviewer for his/her insightful comments in both the revision rounds. The authors are aware of the recent publication by Ferroni and Hopkirk [redacted]. We will include some brief comments on the paper by Ferroni and Hopkirk in our paper here as well [redacted].

Reviewer #2 (Remarks to the Author):

Manuscript is much improved and authors have addressed most comments, however there are still some issues that need to be addressed in the attached file.

Page 3, line 71: "Still, especially for energy pay-back time (which is calculated from cumulative energy demand)". In order to calculate EPBT from CED [MJ/Wp] we need the module (or system) efficiency.

In this case we have calculated EPBT from system CED by dividing the CED/Wp for a complete system by the primary energy equivalent production of 1 Wp of PV system, based on an irradiance of 1700 kWh/m² and a performance ratio of 0.75.

Unfortunately, this does not solve the issue. The CED/Wp does not tell us how much area the panel covers. In order to know this, you need the efficiency at standard irradiance, 1000 W/m². Secondly, the irradiance is incoming solar energy, not how much is converted into electricity for which again you need the panel efficiency. When using the CED/Wp to obtain electricity output, you MUST have either

the capital factor or the panel efficiency. There is no way around this.

To the authors' knowledge it is common practice to calculate the energy yield of a PV system by taking the system capacity, and multiplying it with the yearly insolation figure and the PR to account for all yield losses. See for instance Reich et al., 2012 (doi: 10.1002/pip.1219, <http://onlinelibrary.wiley.com/doi/10.1002/pip.1219/abstract>). Here it is implicit that a yearly insolation of 1700 kWh/m² results in an equivalent of 1700 full-load hours, as the rated capacity (W_p) is established at 1000 W/m². The full calculation thus becomes:

$$\text{Yield} = \text{Capacity} * (\text{insolation} / \text{STC-irradiance}) * \text{PR} = 1 W_p * (1700 \text{ kWh/m}^2 / 1 \text{ kW/m}^2) * 0.75.$$

The methods section has been adapted to more clearly state this (eqn 1-3, lines 305-313).

Page 7, Fig.4: It seems a little disingenuous to plot the 'net energy output' in terms of 'primary energy' when the PV systems would not actually produce this amount of energy. At the least, the word 'equivalent' should make it in there somewhere. Additionally, there should be historical data points on there for comparison.

The figures (4 and 5) and their captions have been updated to reflect that indeed we talk about primary energy equivalent energy, not actual energy produced. As these graphs show simulations / model calculations, it is not possible to include historical datapoints in there, as the data plotted is a combination of historical data on installed capacity, modelled energy impact of production, scenario assumptions on the production of electricity from PV and PV degradation.

It's not clear to me why you cannot add historical data to these plots. I really think it would benefit the reader to be able to gauge how well the model reflects what actually happened.

The manuscript has been updated to include a comparison between the electricity production, installed capacity and global average annual yield based on different datasets: the EIA and UN databases, the data sources for cumulative capacity used for our analysis, and the estimations from the two performance scenarios we have used for our analysis. These data are presented in Figure 6. The discussion section has been updated with a discussion of the different datasets (lines 225-238).

~~*We feel we cannot present historical data points in these figures, as there are no available historical datapoints for the data presented in Figures 4 and 5, e.g. net (primary equivalent) energy production and especially net GHGe avoidance. Although there is some data (e.g. from UN/EIA databases) on electricity production from PV, there is no historical data on the energy consumption of the PV industry, as this was measured in this case by using LCA studies, combining these in experience curves and calculating energy use from this and the growth of the PV industry. Furthermore, the data from the UN database is limited in time horizon, for most countries data is only available from 2010 onwards. Furthermore the data is extremely inaccurate especially for older years.*~~

Page 8, line 174: “The latter result in very unrealistic values for PV performance.” Not clear why this is the case. Needs justification, or reference to methodology section.

Wording has been altered / expanded to more accurately reflect our rationale. Sometimes the databases cover one or the other (capacity or production), and combining two unaligned databases to calculate yield results in these cases in very high or low figures. Also, the top-down databases often estimate electricity production, based on the (estimate of) installed PV capacity and an (estimated) specific yield figure.

Your justification for not using ‘actual’ data seems weak to me. You are throwing out an “aggregate of more and less accurate data.” for a wholly modeled approach. Why would this be more realistic? You need to at least present the results with the two methods.

We wanted to capture historical developments, including the historical development of performance of PV. The data from for instance the UN databases is of insufficient time horizon (e.g. going back only to 1990 for a handful of countries, and for most countries data is only available from 2010 onwards). Dale and Benson for instance used a single figure to determine energy yield, which did not vary over time like we try to do here. We have added some text to explain (lines 219-221, 393-397).

Page 9, line 183: “new PV capacity (and other renewable electricity sources) are more likely to replace older, fossil fuelled power plants as they are decommissioned, mainly coal fired power plants [1], and thus avoided emissions are larger.” While this may be true, PV electricity doesn’t offset purely on the basis of installation/decommission, but at the time-of-production. As such, PV is most likely to offset natural gas peaker plants or hydro, rather than baseline coal.

We agree with the reviewer. The text has been updated to reflect these considerations.

Added text, “However, considering the timing of electricity production from PV, it will likely replace electricity from more sources of generation than baseline power” contradicts the previous sentence. You can’t have your cake and eat it too!

Agreed. We have changed the wording and added a sentence to show we feel that using the grid average is a reasonable approximation.

Page 10, line 235: “Where needed, conversion to the desired units was performed using harmonisation criteria based on LCA guidelines from [12]: 1) a conversion factor from primary energy to electricity of 0.311; 2) an insolation of 1700 kWh·m⁻²·year; 3) a performance ratio (PR) of 0.75, and 4) a module degradation rate of 0.7%/year” This information would not allow to convert between CED or GHG on a per Wp basis without also knowing the efficiency of the module.

The text has been updated

I don't see any text added. See earlier comment regarding converting between capacity (Wp) and electricity production (kWh)

Please see also our answer to an earlier remark. The methods section has been adapted to more clearly state our approach (eqn 1-3, lines 305-312).

Page 12, line 295: What is 'population weighted insolation'? Why would PV only be installed where there is population? In our experience, for the current market, it is the case that most PV capacity is installed in or near urban areas. So, rather than taking a "area-weighted" insolation (e.g. average over whole country area), we have chosen to take insolation weighted by population density. As an extreme example, Sweden, Norway and Finland have a high population density in the southern parts of the countries at highest insolation but all span a large range of latitude, and thus the area-weighted insolation is much lower. For some countries it might be vice versa.

OK, I guess this makes sense (depending on your definition of 'near') and may make a difference for countries with a large population distribution across their north-south axis. However, I still disagree that historical data could not be used. See earlier comment.

Please see also our response to a previous remark. We have screened the available historical data and deemed it insufficient for our calculations here.

Page 13, line 309: "For instance, we have combined statistics from both the U.S. Energy Information Administration (EIA)[76] and the UN statistics database [69] with the installed capacity figures from the IEA PVPS [4, 8] to calculate country- and year-specific PR figures. The values obtained ranged from under 1% to over 800%." With this approach, (i) you should only make calculations within consistent datasets (since different assumptions across datasets will have a large impact) and (ii) it is unclear how you would account for the insolation which the panels receive and thus the PR value is underdetermined. Alternatively, you could calculate the capacity factor, but would have to adjust the data for growth rates. This issue is discussed in Dale and Benson.

Based on your suggestion we have looked deeper into the approach of Dale and Benson. The problem is indeed that we were using databases that are not aligned, thus resulting in very unrealistic values. If we take for instance all data (capacity and generation) from EIA we can calculate much more reasonable figures. The UN databases, combining generation and installed capacity still result in very unrealistic figures. For both, the our two performance ratio scenarios instead. The irradiation figures we use to calculate yield from PR are for fixed tilt, optimally oriented surfaces.

Yes, the UN database has some garbage in there. I would like to see some discussion of comparison between the two results

Please see also our response to a previous remark. We have screened the available historical data

and deemed it insufficient for our calculations here.

Reviewer #3 (Remarks to the Author):

The authors now recognise some previous work looking at cumulative net GHG and cumulative radiative forcing (CRF), by Emmott and Ravikumar (albeit not the reference I suggested: Environ. Sci. Technol. 2014, 48, 10010–10018). They also separate their work from these articles by focussing on global industry growth over longer time periods, and hence satisfy my main concern regarding novelty. I now recommend publication.

The reference by Ravikumar was erroneously the wrong one (problem with reference manager). We now cite the suggested reference. The authors would like to sincerely thank the reviewer for his/her insightful comments and very much appreciate the comments here and the recommendation for publication in Nature Communications. We have updated our article to include reference to the article the reviewer suggested in the first revision

Additional comments received from editor on September 21st:

Please present the gross electricity output from your model, either in Figure 4a or elsewhere, so it can be compared to empirical data on PV energy production from various literature sources, including IEA database and UN database, each database plotted with different symbols. Please include a quantitative discussion comparing the modelled reality you propose and the historical datasets (including a note on the limitations of the datasets and of the model), and if you wish, discussing the differences of interpretations of the model and databases values for the sake of transparency and completeness.

We have updated the manuscript to include a Figure presenting the requested data, and a discussion of the data from the different datasets. Please see an earlier comment as well.

Please clarify which studies were used and which values were used in your meta-analysis and please explain why, for the sake of completeness and transparency. It would be good to discuss and compare the values you use with the literature; for example, the following study has very high values of embodied energy. While this study in particular is just an example, placing your work in the context of previous studies will give it more impact [Mathur, Jyotirmay, Narendra Kumar Bansal, and Hermann-Joseph Wagner. "Dynamic energy analysis to assess maximum growth rates in developing power generation capacity: case study of India." Energy Policy 32.2 (2004): 281-287]

The tables in the Tables section show the studies that we have analyzed, harmonized and filtered to use for our learning curve analysis. It also includes a statement on the filter procedure: e.g. we have only used studies that are for complete PV systems, and have excluded studies that refer to prospective PV system, or studies that refer to worst- or best-case scenarios. A short comment on this procedure was added to section IV.A, on lines (273-275). We have furthermore reviewed the study mentioned in the comment above. The study mentions that it uses data from two studies, one of which is also cited in our manuscript, and in fact used in our analysis (Gürzenich et al, 1999, ref number 63 in our manuscript). The results presented in this study refer to data for PV systems from 1997, and for this year the data is not particularly high compared to the data we present and have used, as can be seen in Table II, 6th data row. The distinction of our analysis compared to that for instance by Mathur et al. is that we use the learning curve approach to account for the development over time (decrease) of environmental impact. Although the study covers a similar topic, e.g. what industry growth rate is sustainable considering the energy footprint of PV systems, it is still based on the assumption that the energy footprint of PV is static, and not developing over time. We have added a short discussion/comparison with this study to the paper at lines (59-62).

Reviewer #2 (Remarks to the Author)

All good. Thank you for the changes.